# Anomaly Detection in Financial Time Series by Principal Component Analysis and Neural Networks †

**Stéphane Crépey** [1,*]**, Noureddine Lehdili** [2]**, Nisrine Madhar** [1,*] **and Maud Thomas** [3]

1. CNRS, Laboratoire de Probabilités, Statistique et Modélisation (LPSM), Team MathFiPronum, Université Paris Cité, 75013 Paris, France
2. Natixis Entreprise Risk Management Department, 75013 Paris, France
3. CNRS, Laboratoire de Probabilités, Statistique et Modélisation (LPSM), Team Statistics, Sorbonne Université, 75005 Paris, France
* Correspondence: stephane.crepey@lpsm.paris (S.C.); madhar@lpsm.paris (M.N.)
† Python notebooks reproducing the results of this paper are available on https://github.com/MadharNisrine/PCANN, (accessed on 12 October 2022). The authors would like to thank Pascal Oswald, Leader Market & Counterparty Risks Modelling, from Natixis, for insightful discussions.

**Abstract:** A major concern when dealing with financial time series involving a wide variety of market risk factors is the presence of anomalies. These induce a miscalibration of the models used to quantify and manage risk, resulting in potential erroneous risk measures. We propose an approach that aims to improve anomaly detection in financial time series, overcoming most of the inherent difficulties. Valuable features are extracted from the time series by compressing and reconstructing the data through principal component analysis. We then define an anomaly score using a feedforward neural network. A time series is considered to be contaminated when its anomaly score exceeds a given cutoff value. This cutoff value is not a hand-set parameter but rather is calibrated as a neural network parameter throughout the minimization of a customized loss function. The efficiency of the proposed approach compared to several well-known anomaly detection algorithms is numerically demonstrated on both synthetic and real data sets, with high and stable performance being achieved with the PCA NN approach. We show that value-at-risk estimation errors are reduced when the proposed anomaly detection model is used with a basic imputation approach to correct the anomaly.

**Keywords:** anomaly detection; financial time series; principal component analysis; neural network; missing data; market risk; value at risk





## 1. Introduction

In the context of financial risk management, financial risk models are of utmost importance in order to quantify and manage financial risk. Their outputs and risk measurements can both help in the process of decision making and ensure that the regulatory requirements are met [1]. Financial management thus heavily relies on financial risk models and the interpretation of their outputs. The data usually consist of time series representing a wide variety of market risk factors. A major issue of such data is the presence of anomalies. A value of the time series is considered to be abnormal whenever its behaviour is significantly different from the behaviour of the rest of the time series [2]. In this work, we focus on the detection of abnormal observations in a market risk factor time series used to calibrate financial risk models. Indeed, erroneous input data may wrongly impact the calibrated model parameters. A typical example is the estimation of the covariance matrix of a bank's market risk factors. This covariance matrix is involved in the computation of the value-at-risk (VaR) or expected shortfalls (expected losses above the VaR level). Since the true covariance is unknown, it has to be estimated from the data. However, the presence of anomalies in the data might have an impact on this estimation. For this specific case, robust

methods less sensitive to anomalies can be used. Yet, existing robust estimators are computationally expensive, with a polynomial or even exponential time complexity in terms of the number of market risk factors. A faster approach was suggested by Cheng et al. [3], but this algorithm only applies to the estimation of the covariance matrix in the case of a Gaussian distribution. Financial risk models used by banks are widespread and various. Therefore, instead of searching for a robust version for each of them, we propose to detect anomalies directly in the time series.

### 1.1. Contributions of the Paper

Our PCA NN methodology identifies anomalies in time series using a principal component analysis (PCA) and neural networks (NN) as the underlying models. This methodology overcomes the common pitfalls associated with anomaly detection. PCA is first used as a feature extractor on the (augmented if needed) data. Anomaly detection is then performed in two steps. The first step identifies the time series with anomalies evaluating the propensity of the time series to be contaminated, as reflected in its so-called anomaly score. Toward this end, we calibrate a feedforward neural network through the minimization of a customized loss. This customized loss allows us to calibrate the cutoff value of the anomaly scores without resorting to expert judgement. In this way, we remove the expert bias. The second step localizes the anomaly among the observed values of the identified contaminated time series.

### 1.2. Outline

The PCA NN approach is detailed in Section 2. Section 3 describes the methodology used for generating the data on which our approach is thoroughly tested in Sections 4–6, exploiting the knowledge of the data-generating process for benchmarking purposes. Section 7 illustrates the benefits of using our approach through a data-cleaning pre-processing stage on a downstream task, namely value-at-risk computations. This is followed by further numerical experiments on real data sets in Section 8. Section 9 concludes the paper. Appendixes A and B provide reviews of the anomaly detection literature and algorithms. Appendix C addresses the data stationarity issue.

## 2. The PCA NN Anomaly Detection Approach

### 2.1. Notations

Let

$$X = \left( X^1, X^2, \ldots, X^i, \ldots, X^n \right)^\top, \tag{1}$$

where the column vector $X^i = \left( x^i_{t_1}, x^i_{t_2}, \ldots, x^i_{t_j}, \ldots, x^i_{t_p} \right) \in \mathbb{R}^p$ corresponds to the $i$-th observed time series, and $x^i_{t_j}$ corresponds to the value observed at time $t_j$ in the $i$-th time series.

Our method fits into the supervised framework. We thus assume that the data matrix $X$ comes with two label vectors. The first label vector $A = \left( A^1, \ldots, A^n \right) \in \{0, 1\}^n$ identifies the time series containing the anomalies, referred to as the contaminated time series. For $i = 1, \ldots, n$, we define the identification labels as

$$A^i = \begin{cases} 1 & \text{if there exists } j \text{ such that } x^i_{t_j} \text{ is an anomaly.} \\ 0 & \text{otherwise.} \end{cases} \tag{2}$$

The second label vector $L$ concerns solely the contaminated time series. Its coefficients correspond to the localization labels, i.e., the time stamps at which an anomaly occurs. For $j = 1, \ldots, p$ and $i \in I_c = \{i : A^i = 1\}$ (the set of contaminated time series),

$$L^i = j \quad \text{when the anomaly occurs at time } t_j \text{ for the } i\text{-th contaminated time series,}$$

$X, A$, and $L$ are general notations. As the model involves a learning phase, we use $X^{Train}$, $A^{Train}$, and $L^{Train}$ to denote the data used for the calibration and $X^{Test}$, $A^{Test}$, and $L^{Test}$ to denote the independent data sets on which the model performance is evaluated.

### 2.2. Methodology Overview

The anomaly detection model we propose is a two-step supervised learning approach. The contaminated time-series identification step aims to identify those with potential anomalies among the observed time series. The anomaly localization step consists of finding the location of the anomaly in each contaminated time series identified during the first step.

The model used in the first step falls under the scope of a binary classification model, assigning to each time series a predicted label

$$\widehat{A}^i = \begin{cases} 1 & \text{if } X^i \text{ is considered contaminated by the identification model.} \\ 0 & \text{otherwise.} \end{cases}$$

The second step uses a multi-classification model. For each contaminated time series identified in the first step, the model predicts a unique time stamp $\widehat{L}^i$ at which the anomaly occurred. Formally, the observed value at the time stamp $t_{\hat{j}^i}$ of the $i$-th contaminated time series is considered abnormal, meaning that the observation $x^i_{t_{\hat{j}^i}}$ is abnormal. Thus,

$$\widehat{L}^i = \hat{j}^i \quad \text{with} \quad \hat{j}^i \in \{1, \ldots, p\}.$$

Our two-step anomaly detection model localizes only one anomaly per run, if any. Several iterations allow for the removal of all anomalies. Indeed, as long as the time series is identified as contaminated in the first step of the method, the time series can proceed to the second step. Once all anomalies have been localized, the final stage of the approach is to remove the abnormal values and suggest imputation values (see Section 6.1).

### 2.3. Theoretical Basics of Principal Component Analysis

A fundamental intuition behind our anomaly detection algorithm is the existence of a lower dimensional subspace, where normal and abnormal observations are easily distinguishable. Since the selected principal components explain a given level of the variance of the data, they represent the common and essential characteristics of all observations. As one might reasonably expect, these characteristics mostly represent the normal observations. When the inverse transformation is applied, the model successfully reconstructs the normal observations with the help of the extracted characteristics, although it fails at reconstructing the anomalies.

In this section, we recall the theoretical tenets of principal component analysis (PCA) that are useful for our purpose. The idea of PCA is to project the data from the observation space of dimension $p$ into a latent space of dimension $k$, with $k < p$. This latent space is generated by the $k$ directions for which the variance retained under projection is maximal.

The first principal component $U$ is given by $U = w^\top X$, where $w$ is a to-be-determined vector of weights. Since PCA aims to maximize the variance along its principal components, $w$ is searched for as

$$\arg\max_{\|w\|=1} w^\top \Sigma w,$$

where $\Sigma$ is the covariance matrix of $X$. The optimal solution by the Lagrangian method is given by $\Sigma w = \lambda w$, where $\lambda > 0$ is a Lagrangian multiplier. Multiplying both sides by $w^\top$ results in $w^\top \Sigma w = \lambda$. Following the same process with an additional constraint regarding the orthogonality between the principal components, one can show that the $k$ first principal components correspond to the eigenvectors associated with the $k$ dominant eigenvalues of $\Sigma$.

The transfer matrix $\Omega_X \in \mathbb{R}^{k \times p}$ is defined by the eigenvectors of $\Sigma$ associated with its $k$ largest eigenvalues. This transfer matrix $\Omega_X$ applied to any observation in the original

observation space projects it into the latent space. Since the transformation is linear, reconstructing the initial observations from their equivalent in the latent space is straightforward. Using PCA on the data matrix $X$, the transfer matrix $\Omega_X$ can be inferred. The projection of the observations into the latent space $Z \in \mathbb{R}^{n \times k}$ is then given by $Z = X\Omega_X^\top$, whereas their reconstructed values are given by $\widehat{X} = Z\Omega_X$.

The reconstruction errors $\varepsilon^i$ for each time series defined by

$$\varepsilon^i = \widehat{X}^i - X^i, \quad i = 1, \ldots, n, \tag{3}$$

referred to in the sequel as the new features, are our new representation of data, given as inputs to our models. Algorithm 1 summarizes how to compute these reconstruction errors. The two steps of the model use the same inputs, namely these reconstruction errors. Nevertheless, the processing these inputs undergo in each step differs.

---

**Algorithm 1** Feature extraction with PCA

---

**inputs:** number of principal components $k$
out of sample data $X^{Test}$
training data if not trained yet $X^{Train}$
**if** *not trained yet* **then**
$\quad \mid \quad \Omega_{X^{Train}} \leftarrow \text{PCA}(X^{Train}, k)$
**end**
$\varepsilon^{Train} = X^{Train}\left(\Omega_{X^{Train}}^\top \Omega_{X^{Train}} - I_p\right)$
$\varepsilon^{Test} = X^{Test}\left(\Omega_{X^{Train}}^\top \Omega_{X^{Train}} - I_p\right)$

**outputs:** $\varepsilon^{Train}$ reconstruction errors for the train set
$\quad\quad\quad \varepsilon^{Test}$ reconstruction errors for the test set

---

### 2.4. Contaminated Time Series Identification

The first step of the approach aims to identify the time series with anomalies based on the reconstruction errors obtained using PCA and Algorithm 1. In this section, we detail the process these new features undergo. We assign an anomaly score to each reconstruction error, whose computation is handled by a neural network (NN) (see [4]). We also numerically demonstrate that our NN has an advantage over a naive approach in terms of being able to assess more accurately the propensity of a time series to be contaminated.

#### 2.4.1. Naive Approach

A naive and natural approach to defining the anomaly score is to consider the $\ell^2$-norm of the reconstruction errors $\widetilde{\varepsilon}_i = \|\varepsilon^i\|_2, i = 1, \ldots, n$. This anomaly score represents the propensity of a time series to be contaminated. The scores are first split into two sets depending on whether or not the corresponding time series is contaminated. We use $\widetilde{\varepsilon}^c$ and $\widetilde{\varepsilon}^u$ to denote the set of contaminated and uncontaminated time series anomaly scores, i.e.,

$$\widetilde{\varepsilon}^c := \{\|\varepsilon^i\|_2; \ A^i = 1\} = \{\widetilde{\varepsilon}_i, \ i \in I_c\}$$
$$\widetilde{\varepsilon}^u := \{\|\varepsilon^i\|_2; \ A^i = 0\} = \{\widetilde{\varepsilon}_i, \ i \notin I_c\}.$$

The density distribution function of each class of time series $f^u$ and $f^c$, i.e., for both $\widetilde{\varepsilon}^c$ and $\widetilde{\varepsilon}^u$, is then estimated using the kernel density estimation [5]. For $l = \{c, u\}$, let $\varepsilon^l$ be an i.i.d sample of $n^l$ observations from a population with an unknown density $f^l$. The corresponding kernel estimator is given by

$$\hat{f}^l(s) = \frac{1}{n^l \mathfrak{h}} \sum_{k=1}^{n^l} \mathcal{K}\left(\frac{s - \widetilde{\varepsilon}_k^l}{\mathfrak{h}}\right), \tag{4}$$

where $\mathcal{K}$ is a kernel function and $\mathfrak{h}$ is a smoothing parameter.

The cutoff value $s$ is then chosen as the intersection between the two empirical density functions, i.e.,

$$\hat{s} = \arg\min_s \left\{ |\hat{f}^u(s) - \hat{f}^c(s)| < \eta \right\},$$

where $\eta$ is a to-be-tuned precision level. The selected cutoff value $\hat{s}$ represents the value of the score for which the area under the curve of the density of uncontaminated time series above $\hat{s}$ and the area under the curve of contaminated time series below $\hat{s}$ are as small as possible. We expect it to correspond to a value exceeded by a relative low number of $\widetilde{\varepsilon}_i$ with $i \notin I_c$ as $\hat{f}^u(s)$ is expected to be flat around the cutoff value exceedance region and exceeded only by a few $\widetilde{\varepsilon}_i$ with $i \in I_c$.

As shown in Figure 1, the naive approach results in a non-negligible overlapping region between the two densities, for both the training set (left) and test set (right). This region represents the anomaly score associated with time series that could be either contaminated or uncontaminated. The uncertainty regarding the nature of the observation when its anomaly score belongs to this region is relatively high in comparison with the observations whose anomaly scores lie on the extreme left-hand or right-hand side of the calibrated cutoff value. Moreover, because we are not able to provide a clear separation between the scores of uncontaminated and contaminated time series, we may expect the model to mislabel future observations, resulting in a high rate of false positives and true negatives.

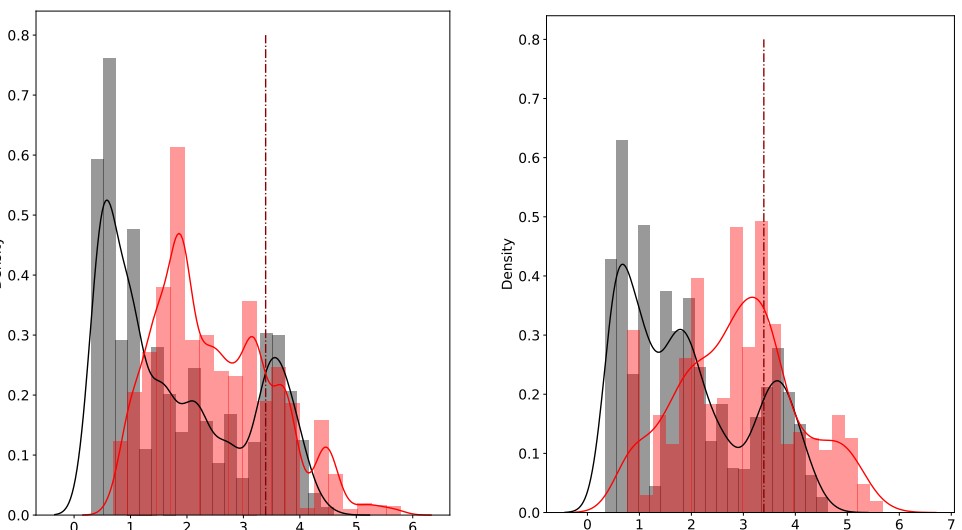

**Figure 1.** Empirical densities of anomaly scores given by the naive approach for uncontaminated time series in black and contaminated time series in red in a training set (**left**) and test set (**right**). The dotted dark red lines represent the cutoff values.

### 2.4.2. Neural Network Approach

In view of obtaining a clearer separation of the densities, we propose an alternative approach for the computation of the anomaly scores built upon the reconstruction errors. Let $F$ denote the function that associates an anomaly score with each reconstruction error $\varepsilon^i$. In the naive approach, $F$ corresponds to the $\ell^2$-norm. Hereafter, $F$ instead represents the outputs of a feedforward neural network (NN). The training of the corresponding NN aims to minimize a loss function that reflects our ambition to construct a function $F$, giving accurate anomaly scores. The network used for computing the anomaly scores is defined by

$$F(\varepsilon) = \left( h^H \circ h^{H-1} \circ \ldots h^2 \circ h^1 \right)(\varepsilon),$$

where $h^i(\varepsilon) = \left( W^i \cdot \varepsilon + b^i \right)^+$ represents the computation carried out in the $i$-th layer of the neural network with $n_{h^i}$ hidden units. $H$ stands for the number of layers of the NN, with

ReLU activation applied element-wise. We finally estimate the labels from the outputs following $\hat{A} = \mathbb{1}_{\{(F(\varepsilon)-s)^+>0\}}$.

The idea of the learning is to calibrate the weights $W := \left\{ W^i \in \mathbb{R}^{n_h^i \times n_h^{i-1}}, i = 2, \ldots, H \right\}$ and the biases $b := \left\{ b^i \in \mathbb{R}^{n_h^i}, i = 1, \ldots, H \right\}$ so that the NN is able to accurately assess the extent to which a time series is contaminated, with a clear distinction between the anomaly scores assigned to uncontaminated time series and those assigned to contaminated ones while integrating the calibration of the cutoff value $s$ as part of the learning. We use $\Theta := \{W, b, s\}$ to denote the set of parameters to be calibrated through the NN training. To meet these needs, the loss we minimize during the learning is given by

$$\mathcal{L}_{\Theta}(A, \hat{A}) = \text{BCE}(A, \hat{A}) + \text{AUCDensity}^u_{A, F(\varepsilon)} + \text{AUCDensity}^c_{A, F(\varepsilon)}, \tag{5}$$

where

$$BCE(A, \hat{A}) = -\frac{1}{n} \sum_i \left( A^i \log(\hat{A}^i) + (1 - A^i) \log(1 - \hat{A}^i) \right)$$

is the binary cross-entropy, a well-known loss function classically used for classification problems. To this first component of our loss function, we add two components that aim to downsize the overlapping region between the densities of the anomaly scores of both types of observations. In order to have control of the latter, we consider AUCDensity$^u$ and AUCDentsity$^c$, which correspond to the area under the curve of the probability density function of the anomaly scores the model assigns to contaminated and uncontaminated observations, respectively, i.e.,

$$\text{AUCDensity}^u(s) = \int_s^\infty \hat{f}^u_{A, F(\varepsilon)}(\omega) d\omega , \qquad \text{AUCDensity}^c(s) = \int_{-\infty}^s \hat{f}^c_{A, F(\varepsilon)}(\omega) d\omega . \tag{6}$$

The bounds of these integrals depend on the cutoff value $s$ and define the region for which we want the probability density functions to be as small as possible, which allows us to estimate $\hat{s}$. The estimated probability density functions $\hat{f}^u_{A, F(\varepsilon)}$ and $\hat{f}^c_{A, F(\varepsilon)}$ depend on $F(\varepsilon)$, the scores assigned by the contaminated time series identification model, and the identification labels $A$. In Algorithm 2, we describe the scoring and cutoff value calibration achieved by the calibration of a feedforward network with the loss (5). Each update of $\Theta$ AdamStep in Algorithm 2 is carried out following the Adam optimization algorithm of [6].

Figure 2 and Table 1 show that with the NN approach, the densities of the scores assigned to each type of time series display the expected behaviours on the left-hand (right-hand) side of the cutoff value for contaminated (uncontaminated) time series for both the training and test sets. Actually, the lower the AUCDensity$^u$ (AUCDensity$^c$), the lower the number of uncontaminated (contaminated) time series to which the anomaly scores above (below) the cutoff value are assigned, which prevents mislabelling.

**Table 1.** AUC obtained with the naive and the NN approaches for the training set (**left**) and the test set (**right**).

| Approach | AUCDensity$^u$ | AUCDensity$^c$ | Approach | AUCDensity$^u$ | AUCDensity$^c$ |
|----------|----------------|----------------|----------|----------------|----------------|
| Naive | 0.1725 | 0.7898 | Naive | 0.1897 | 0.6354 |
| NN | 0.05153 | 0.1550 | NN | 0.1290 | 0.1367 |

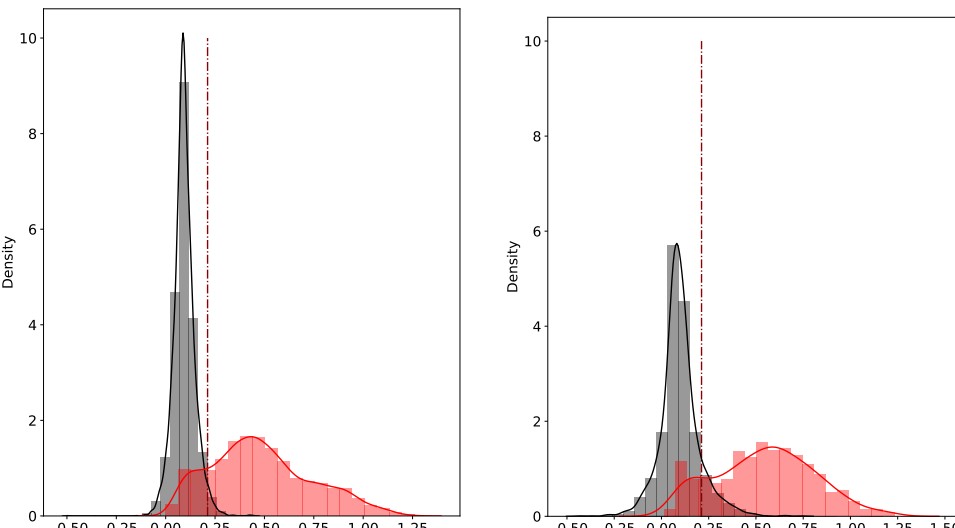

**Figure 2.** Empirical densities of anomaly scores given by the NN approach for uncontaminated time series in black and contaminated time series in red for the training set (**left**) and test set (**right**). The dotted dark red lines represent the calibrated cutoff value $\hat{s}$.

---

**Algorithm 2** Scoring and cutoff calibration

---

**inputs:** learning rate $lr$
    number of maximum iterations $K$
    kernel density estimator parameters $\mathcal{K}, \mathfrak{h}$
    training data $\varepsilon^{Train}, A^{Train}$
Initialize parameter $\Theta$, $\widehat{\mathcal{L}} = \infty$ and count index $k = 0$
**while** $k < K$ **do**

   $scores \leftarrow F_\Theta\big(\varepsilon^{Train}\big)$
   $scores^c \leftarrow scores\mathbb{1}_{A^{Train}=1}$
   $scores^u \leftarrow scores\mathbb{1}_{A^{Train}=0}$         `// Density estimation following (4)`
   $\hat{f}^u_{A^{Train},scores^u} = \text{KernelDensityEstimator}(\mathcal{K}, \mathfrak{h}, scores^u)$
   $\hat{f}^c_{A^{Train},scores^c} = \text{KernelDensityEstimator}(\mathcal{K}, \mathfrak{h}, scores^c)$
   `// Loss evaluation following (5) and (6)`
   $\text{AUCDensity}^u \leftarrow \text{NumericalIntegration}(\hat{f}^u_{A^{Train},scores^u}, s)$
   $\text{AUCDensity}^c \leftarrow \text{NumericalIntegration}(\hat{f}^c_{A^{Train},scores^c}, s)$
   $\hat{A} \leftarrow \mathbb{1}_{scores>s}$
   $\mathcal{L} \leftarrow \text{BCE}\big(A^{Train}, \hat{A}\big) + \text{AUCDensity}^u + \text{AUCDensity}^c$
   **if** $\widehat{\mathcal{L}} > \mathcal{L}$ **then**
     $\widehat{\mathcal{L}} \leftarrow \mathcal{L}$
     $\widehat{\Theta} \leftarrow \Theta$
   **end**

   $\Theta \leftarrow \text{AdamStep}(\mathcal{L}, \Theta, lr)$
   $k \leftarrow k + 1$
**end**
**outputs:** best calibrated parameter $\widehat{\Theta} = \{\widehat{W}, \hat{b}, \hat{s}\}$.

---

Once the features $\varepsilon$ and the calibrated NN are provided, the contaminated times series identification model is ready for use. This step is described by Algorithm 3.

---

**Algorithm 3** Contaminated time series identification model

---

**inputs:** time series to analyze $X^i$,

      calibrated model parameter $\widehat{\Theta}$,

$\varepsilon^i \leftarrow$ PCAFeaturesExtraction$(X^i)$ ;            `// cf. Algorithm 1 in Section 2.3`

$score^i \leftarrow F_{\widehat{\Theta}}(\varepsilon^i)$

$\hat{A}^i \leftarrow \mathbb{1}_{score^i > \hat{s}}$

**outputs:** identification label $\hat{A}^i$.

---

### 2.5. Anomaly Localization Step

Once the time series containing anomalies have been identified, the second step of our approach aims to localize an abnormal observation among each contaminated time series. Again, we use the reconstruction errors defined in (3) as the inputs of this second step. The difference lies in the transformation these reconstruction errors undergo before labelling the different time stamp observations. We now consider the following transformation of the reconstruction errors:

$$F(\varepsilon^i) = |\varepsilon^i| = |X^i - \widehat{X}^i|,$$

with $|\cdot|$ meaning component wise. The model input $F(\varepsilon^i) \in \mathbb{R}^p_+$ is thus assigned to each time series $X^i \in \mathbb{R}^p$. The time stamp of the occurrence of the anomaly is given by

$$\widehat{L}^i := \arg\max_j \left\{ F(\varepsilon^i_{t_j}) : j \in \{1, \dots, p\} \right\}.$$

Algorithm 4 recaps the anomaly localization model.

---

**Algorithm 4** Anomaly Localization Model

---

**inputs:** time series $X^i$ to analyze.

$\varepsilon^i \leftarrow$ PCAFeaturesExtraction$(X^i)$

$\widehat{L}^i \leftarrow \arg\max_j \left\{ |\varepsilon^i_{t_j}| : j \in \{1, \dots, p\} \right\}$

**outputs:** anomaly location $\widehat{L}^i$.

---

Note that anomalies are not necessarily extrema. For this reason, the above PCA feature extraction step is necessary. Building on the reconstruction errors, the model identifies these extrema anomalies and also abnormal observations that are not necessarily extrema. This subtlety of the nature of the observations underlines the importance of going further than taking the index of the highest observed value of the contaminated time series $X^i$.

The flow chart in Figure 3, along with Algorithm 5, summarize our approach. When a time series $X$ is given to our model, we suggest a new representation of $X$, namely $\varepsilon$, through a feature engineering step involving PCA. The resulting representation $\varepsilon^i$ feeds the first component of the model, i.e., the identification step, which evaluates the time series propensity to be contaminated. The optimal parameter $\widehat{\Theta}$ of the identification model solves

$$\widehat{\Theta} = \arg\min_{\Theta=(W,b,s)} \sum_{X \in \boldsymbol{X}} \mathcal{L}_{\Theta}(A, \hat{A}),$$

where $\hat{A} = \mathbb{1}_{F\left((\Omega\Omega^\top - I_p)X\right) > s}$ and $\mathcal{L}_{\Theta}$ is the loss function defined in (5). Then, if the model considers the time series contaminated, the localization model takes over to localize the abnormal value in $X$. For this second step of the model, the time stamp of the occurrence of the anomaly is given by

$$\arg\max_j \left\{ \left( X\left(\Omega^\top\Omega - I_p\right) \right)_j : j \in \{1, \dots, p\} \right\}$$

Finally, the anomaly is imputed. As our approach integrates the computation of a reconstruction of the time series, we could replace the anomaly with the corresponding reconstructed value. The model will be then able to detect the anomaly and suggest an imputation value. However, the numerical tests described in Section 6.1 show that imputations with naive approaches perform better.

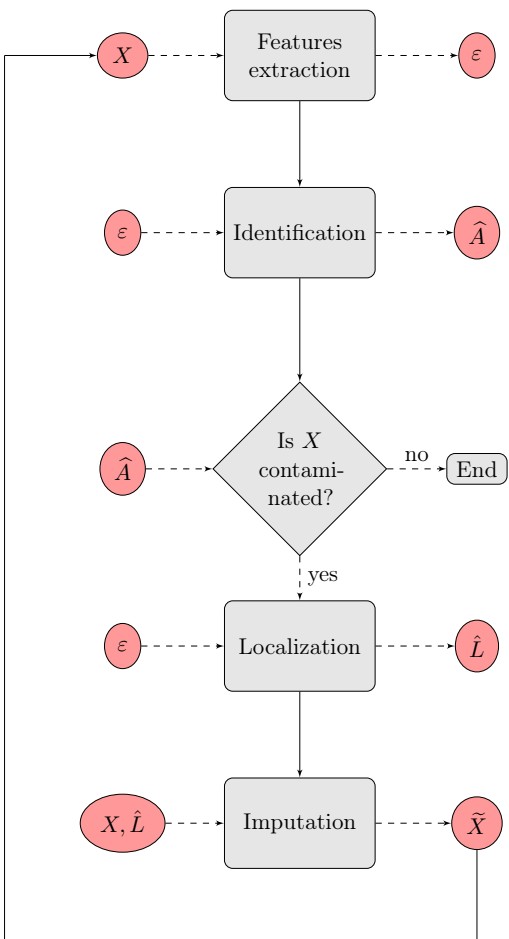

**Figure 3.** Flow chart of our two-step anomaly detection model (PCA NN), depicting the process a time series *X* undergoes.

---

**Algorithm 5** Anomaly Detection Model

---

**inputs:** time series $X^i$ to analyze,

        calibrated identification model parameter $\widehat{\Theta}$

$\widehat{A}^i \leftarrow \text{IdentificationModel}(X^i, \widehat{\Theta})$                  `// Algorithm 3`

  **if** $\widehat{A}^i = 1$ **then**

  │   $\widehat{L}^i \leftarrow \text{LocalizationModel}(X^i)$               `// Algorithm 4`

**end**

**outputs:** identification label $\widehat{A}^i$,

        anomaly localization $\widehat{L}^i$.

---

## 3. Data Generation Process

Since anomalies are rare by definition, real data sets are very imbalanced; the proportion of anomalies compared to normal observations is very small, making the learning phase of the model difficult. This led us to first consider synthetic data for the model calibration. The corresponding data set is obtained through a three-step process including

time series simulations, contamination, and data augmentation. We should point out that in the data simulation, care was taken to ensure that the generated data sets stayed realistic. In particular, only a few anomalies were added to the time series as described in Section 3.2. To this extent, we still faced the problem of the scarcity of anomalies in this synthetic framework, and we provided some pre-processing steps to sidestep this issue.

### 3.1. Data Simulation

In this section, we describe the model used to simulate the data. Recall that our primary motivation is to detect anomalies in financial time series. For that purpose, we consider the share price sample paths generated using the Black and Scholes model, i.e., geometric Brownian motions. Under this framework, the share price $S_t$ is defined by

$$S_t = S_0 \exp\left(\left(\mu - \frac{1}{2}\sigma^2\right)t + \sigma W_t\right). \tag{7}$$

where $W$ represents a standard Brownian motion, $\mu$ the drift, $\sigma$ the volatility of the stock, and $S_0$ the initial stock price.

Let $N$ stocks $S^1, \ldots, S^N$ be simulated simultaneously from this model, where $S^i := \left\{S^i_{t_0}, S^i_{t_1}, \ldots, S^i_{t_T}\right\}$ is the time series of length $T$ representing the (time-discretized) path diffusion of the $i$-th stock. Each stock $S^i$ has its own drift $\mu^i$, volatility $\sigma^i$, and initial value $S^i_{t_0}$. The paths parameters are selected randomly according to

$$S^i_{t_0} \sim \mathcal{N}(100, 1), \qquad \mu^i \sim \mathcal{U}([0.01, 0.2]), \qquad \sigma^i \sim \mathcal{U}([0.01, 0.1]), \quad i = 1, \ldots, N. \tag{8}$$

For the sake of realism, the Brownian motions driving the $N$-stocks are correlated. The contamination of the obtained time series by anomalies is described in the next section.

### 3.2. Time Series Contamination

We apply a quite naive approach to introducing anomalies into our time series. We introduce the same fixed number of anomalies $n^{anom}$ to each time series by applying a shock to some original values of the observed time series. Formally, the $j$-th added anomaly is characterized by its location $t_j$ corresponding to the time stamp at which the anomaly has occurred, its shock $\delta_j$, the amplitude of the shock is given by $|\delta_j|$, and its sign by $\text{sgn}(\delta_j)$. We use $S^{a,i}$ to denote the time series resulting from the contamination of the $i$-th clean time series $S^i$. For $i \in \{1, \ldots, N\}$, let $\mathcal{J}^i$ be the set of indices of the time stamps at which an anomaly occurs for the $i$-th time series. For $j \in \mathcal{J}^i$, the abnormal values are

$$S^{a,i}_{t_j} = S^i_{t_j}\left(1 + \delta_j\right). \tag{9}$$

The location, sign, and amplitude of the shocks are generated randomly according to uniform distributions:

$$\mathcal{J} \sim \mathcal{U}_{N,n^{anom}}(\{1, \ldots, p\}), \qquad \text{sign}(\delta) \sim \mathcal{U}_{N,n^{anom}}(\{-1, 1\}), \qquad |\delta| \sim \mathcal{U}_{N,n^{anom}}([0, \rho]),$$

where $\rho$ is an upper bound on the shock amplitude.

We define the anomaly mask matrix $\mathbb{R}^{N \times T}$ by setting $i = 1, \ldots, N$, and $j = 1, \ldots, T$,

$$\mathcal{T}_{i,j} = \begin{cases} 1 + \delta_j & \text{if } j \in \mathcal{J}^i. \\ 1 & \text{otherwise.} \end{cases} \tag{10}$$

Under this framework, when we incorporate the anomalies into the clean time series driven by the geometric Brownian motion, we assign labels to the time series observations according to whether the values correspond to anomalies or normal observations. We

thus provide the labels $Y_{t_j}^i$ associated with each value $S_{t_j}^{a,i}$. Hence, for $i = 1, \ldots, N$, and $j = 1, \ldots, T$,

$$
Y_{t_j}^i = \begin{cases} 1 & \text{if } j \in \mathcal{J}^i. \\ 0 & \text{otherwise.} \end{cases} \tag{11}
$$

Algorithm 6 recaps the time series contamination procedure, where AnomalyMask and GetLabels refer to the operators defined by (10) and (11).

---

**Algorithm 6** Data Contamination

---

**inputs:** amplitude range $\rho$,
         number of anomalies to add in time series $n^{anom}$,
         data $S$
$\text{sgn}(\delta) \leftarrow \mathcal{U}_{N,n^{anom}}(\{-1,1\})$
$|\delta| \leftarrow \mathcal{U}_{N,n^{anom}}(\{0,\rho\})$
$\mathcal{J} \leftarrow \mathcal{U}_{N,n^{anom}}(\{1,\ldots,p\})$
$\mathbb{R}^{N \times T} \ni \mathcal{T} \leftarrow \text{AnomalyMask}(\mathcal{J}, \delta)$
$S^a \leftarrow S \circ \mathcal{T}$
$Y \leftarrow \text{GetLabels}(\mathcal{J})$
**outputs:** time series with anomalies $S^a$,
            labels associated with each value of the time series $Y$.

---

### 3.3. Data Augmentation

By definition, anomalies are rare events and thus represent only a low fraction of the data set. Yet, the suggested approach needs an important training set for efficient learning. To overcome this issue, we apply a sliding window data augmentation technique [7]. This method not only extends the number of anomalies within the data set but also allows the model to learn that anomalies could be located anywhere in the time series. It consists of extracting $N_p = T - p + 1$ sub-time series of length $p$ of the initial observed time series of length $T$.

Note that we have to split the data into training and test sets before augmentation to guarantee that the time series considered in the training set do not share any observations with the ones we used for the model evaluation. In view of simplification, we introduce the data augmentation process without a loss of generality for $S$ and $Y$. In practice, this process has to be applied to $S^{Train}$ and $Y^{Train}$, and $S^{Test}$ and $Y^{Test}$ separately.

For $i = 1, \ldots, N$ and $q \in \{1, \ldots, N_p\}$, the sub-time series $S^{i,q}$ and the associated labels $Y^{i,q}$ are defined by Algorithm 7

$$
S^{i,q} = \left\{ S_{t_j}^i, \ j = q, \ldots, q + p - 1 \right\}
$$
$$
Y^{i,q} = \left\{ Y_{t_j}^i, \ j = q, \ldots, q + p - 1 \right\}.
$$

Each sub-time series $S^{i,q+1}$ thus results from the shift forward in time of one observation of the previous sub-time series $S^{i,q}$. The final data set $X$ and the labels $^sY$ are then defined as the matrices whose rows correspond to the sub-time series $S^{i,q}_{q=1,\ldots,N_p;i=1,\ldots,N}$, and $Y^{i,q}_{q=1,\ldots,N_p;i=1,\ldots,N}$, respectively, i.e.,

$$
X = \left( S^{1,1}, S^{1,2}, \ldots, S^{1,N_p}, S^{2,1}, \ldots, S^{i,q}, \ldots, S^{N,N_p} \right)^\top,
$$
$$
^sY = \left( Y^{1,1}, Y^{1,2}, \ldots, Y^{1,N_p}, Y^{2,1}, \ldots, Y^{i,q}, \ldots, Y^{N,N_p} \right)^\top.
$$

With this data configuration, an observation refers to a time series $S^{i,q}$ obtained through the sliding window technique. Two observations may represent the same stock but at different time intervals.

---

**Algorithm 7** Data Augmentation with sliding window technique

---

**inputs:** window size $p$,

        data and labels $S,Y$;

 Initialize empty slided time series and labels matrices $X$, $^sY$;

 **for** $i :=1$ *to* $N$ **do**

    **for** $q :=1$ *to* $N_p$ **do**

        $S^{i,q} \leftarrow \left\{ S^i_{t_j} | j \in \{q,\dots,q+p-1\} \right\}$

        $Y^{i,q} \leftarrow \left\{ Y^i_{t_j} | j \in \{q,\dots,q+p-1\} \right\}$

    **end**

    $X \leftarrow \text{Concatenate}\left(X, S^{i,q}\right)$

    $^sY \leftarrow \text{Concatenate}\left(^sY, Y^{i,q}\right)$

**end**

**outputs:** resulting slided time series and labels $X, {}^sY$

---

Although the fact that the observations share the same values can be argued to wrongly impact the learning process, we should point out that a real benefit can be drawn from this situation. Indeed, thanks to this sliding window technique, the number of anomalies is considerably increased. This technique also allows the model to learn that anomalies could be located anywhere in the time series, reducing the dependency on the event location [8].

However, once we apply the sliding window technique, we do not only extend the number of contaminated time series; the number of uncontaminated time series is also increased. However, the minority class (contaminated time series) has at least a significant number of instances, denoted by $N^c$. In order to obtain a more balanced data set for the training set, we perform an undersampling, randomly selecting $N^c$ observations from the $N^u$ uncontaminated time series without any anomalies (RandomSampling in Algorithm 8). The resulting retained number of observations $2N^c$ is now sufficient to train the model. The test set in turn is imbalanced. To sharpen the imbalanced characteristic of the test set, we specify a contamination rate $r_c$, which corresponds to the rate of contaminated time series in the data set. The construction of the test set is described in Algorithm 8.

As mentioned in Section 2.2, Algorithm 5 is designed to predict the localization of only one anomaly (if there is more than one anomaly in the time series, it should be run iteratively as detailed in Section 2.2). Here, we only consider time series with at most one anomaly (without loss of generality).

**Assumption 1.** *A time series $S^{i,q}$ contains at most one anomaly among all its observed values.*

We assign the identification label $A^{i,q}$ (see Section 2.1) to each $S^{i,q}$ following the rule

$$A^{i,q} = \sum_{j=q}^{q+p-1} Y^i_{t_j} = \begin{cases} 1 & \text{if there is an anomaly among the observed values of } S^{i,q} \\ 0 & \text{otherwise.} \end{cases}$$

Regarding the localization labels, we recall that they only concern time series with an anomaly, therefore $L^{i,q}$ is defined as follow:

$$L^{i,q} = \arg\max_j Y^{i,q} = \arg\max_j \left\{ Y^i_{t_j}, \, j = q,\dots,q+p-1 \right\} \tag{12}$$

With Algorithm 9, we give a rundown of the construction process of the identification and localization labels, namely $A$ and $L$ departing from $^sY$.

---

**Algorithm 8** Time series selection

---

**Inputs:** slided data and labels $X$, $^sY$,
contamination rate $r_c$ (for test set)

$X$, $^sY \leftarrow X\mathbb{1}_{\text{sum}(^sY)\leq1}$, $^sY\mathbb{1}_{\text{sum}(^sY)\leq1}$ ;        `// Keep time series with at most one`
`anomaly.`
**if** *Train set* **then**

  $N^c = \text{Card}\left(^sY\mathbb{1}_{\text{sum}(^sY)=1}\right)$

  `// Randomly select the indexes of` $N^c$ `uncontaminated time series`
  $\text{index}^u = \text{RandomSampling}\left(\{i;\ \text{sum}(Y_s^i) = 0, i \in \{1,\ldots,NN_p\}\}, N^c\right)$

**end**
**if** *Test set* **then**

  $N^c = \text{Card}\left(^sY\mathbb{1}_{\text{sum}(^sY)=1}\right)$

  $N^u = \left\lceil \frac{N^c(1-r_c)}{r_c} \right\rceil$

  `// Randomly select the indexes of` $N^c$ `uncontaminated time series`
  $\text{index}^u = \text{RandomSampling}\left(\{i;\ \text{sum}(Y_s^i) = 0, i \in \{1,\ldots,NN_p\}\}, N^u\right)$

**end**

 $\text{index}^c = \{i;\ \text{sum}(Y_s^i) = 1, i \in \{1,\ldots,NN_p\}\}$
 $X \leftarrow \left(X^i, \text{for } i \in \text{index}^u \cup \text{index}^c\right)$
 $^sY \leftarrow \left(Y_s^i, \text{for } i \in \text{index}^u \cup \text{index}^c\right)$

**outputs:** time series $X$ with at most one anomaly,
corresponding labels for identification task $A$,
corresponding labels for localization task $L$.

---

The supervised learning framework is adopted herein since we have at our disposal the labelled data.

The resulting matrices with the observed values of the time series in $X$, the associated identification labels in $A$, and the localization labels in $L$ constitute the data set used for our model calibration and evaluation.

---

**Algorithm 9** Time series labelling

---

**Inputs:** slided labels $^sY$;
$A \leftarrow \text{sum}(^sY)$

$L \leftarrow \arg\max\left\{^sY\mathbb{1}_{\text{sum}(^sY)=1}\right\}$

**outputs:** corresponding labels for identification task $A$,
corresponding labels for localization task $L$.

---

## 4. Model Evaluation: Setting the Stage

After introducing the relevant performance metrics used to assess the PCA NN performance, we describe the synthetic data used for the numerical experiments in Sections 5–7. We then briefly describe the process of the latent space dimension calibration.

### 4.1. Performance Metrics

Common methods used to assess the performance of binary classifiers include true positive and true negative rates, as well as ROC (Receiver Operating Characteristics) curves, which display the true positive rate against the false positive rate. These methods, however, are uninformative when the classes are severely imbalanced. In this context, $F_1$-scores and Precision-Recall Curves (PRC) should be used [9,10]. They are both based on the values of

$$\text{Precision}(s) = \frac{\text{true positives}}{\text{true positives} + \text{false positives}}$$

against the values of

$$\text{Recall}(s) = \frac{\text{true positives}}{\text{true positives} + \text{false negatives}}$$

where $s$ is a cutoff probability varying between 0 and 1. Precision quantifies the number of correct positive predictions out of all positive predictions made, whereas recall (often also called sensitivity) quantifies the number of correct positive predictions out of all positive predictions that could have been made. Both focus on the positive class (the minority class, anomalies) and disregard the negative (the majority class, normal observations).

Our $F_1$ score [11,12] combines these two measures in a single index defined as

$$F_1 = 2 \cdot \frac{\text{Precision} \cdot \text{Recall}}{\text{Precision} + \text{Recall}}. \tag{13}$$

The closer the $F_1$-score to 1, the better the prediction model.

A PRC displays the values of precision and recall as the cutoff $s$ varies from 0 to 1. The PRC of a skillful model bows towards the point with coordinates $(1, 1)$. The curve of a no-skill classifier is a horizontal line at some $y$-level proportional to the proportion of positives in the data set. For a balanced data, set this proportion is just 0.5 [9].

The PRC and $F_1$-score are complementary in our approach. The $F_1$-score is used on the anomaly scores outcomes of the models to identify the best configuration and the best model. A PRC is used to select the best cutoff value used in the prediction of the two classes for each model.

### 4.2. Synthetic Data Set

The anomaly detection task is performed on $N = 20$ stocks simultaneously. The stock prices are diffused according to the Black and Scholes model. Each stock has its own drift and volatility and the 20 stocks are correlated, as described in Section 3. Each time series represents $T = 1500$ the daily stock prices, split into two sets. The first 1000 observations corresponding to the training set are used to learn the model parameters, i.e., the PCA transfer matrix and the NN weights and biases. The last 500 observations corresponding to the test set are used to assess the quality of the estimated parameters when applied to unseen samples. For the application of the sliding window technique, we set the length of the resulting time series $p$ to be 206. Table 2 sums up the composition of each data set before and after data augmentation.

**Table 2.** Data set composition (**top**) before and (**bottom**) after data augmentation.

|  | Nb of Time Series | Nb of Observed Values per Time Series | Nb of Anomalies |
|---|---|---|---|
| Train set | 20 | 1000 | $4 \times 20$ |
| Test set | 20 | 500 | $2 \times 20$ |
|  | **Nb of Time Series** | **Nb of Observed Values per Time Series** | **Nb of Anomalies** |
| Train set | 12,000 | 206 | 6000 |
| Test set | 2500 | 206 | 400 |

We recall that both steps of the approach are preceded by a feature extraction step, for which the latent space dimension $k$ needs to be calibrated. As shown by the numerical results provided in Appendix C, the feature extraction also guarantees the stationarity of the time series used for the anomaly detection task, $\varepsilon$.

### 4.3. Calibration of the Latent Space Dimension

When performing PCA, $k$ is determined through a scree plot, which is the representation of the proportion of variance explained by each component. The optimal $k$ corresponds to the number of principal components explaining a given level of the variance of the original data. However, this method has its limitation as stated in [13]. More importantly, it is not suitable for our approach. Indeed, in our case, the number of selected principal components must achieve a trade-off between information and noise in the latent space. If we consider a number of principal components that is too low, we may lose information regarding the normal observations, leading to false alarms. If too high a number of principal components is retained, we may include components representing noise, which prevents the model from detecting some anomalies.

To select the optimal dimension of the latent space, we consider the distribution of anomaly scores obtained through the naive approach in Section 2.4, empirically calibrating a cutoff $\hat{s}$ through the non parametric estimation of the distribution of the anomaly scores of uncontaminated and contaminated time series. We chose to calibrate the dimension of the latent space with the naive approach, because using the NN to this end would be very costly. Hence, for each $k = 5, 10, \ldots, 200$, we construct a PCA model from which we infer reconstruction errors, which are then converted into anomaly scores. Based on these anomaly scores, we tune the cutoff value $\hat{s}$ thanks to the distributions and we finally convert the scores into labels. We evaluate the predictions of the naive approach for each value of $k$. The results of the evaluation metrics on the training set, as legitimate to choose $k$, are represented in Figure 4. The highest values are reached for $k \in \{40, \ldots, 145\}$. The performance seems to be stable in terms of $F_1$-score and accuracy. Therefore, for computational reasons, we choose the optimal $k$ to be 40.

Figure 4 also shows, as expected, a downward trend in the scores when presented against the highest values of $k$. This demonstrates that when a high level of variance is explained, it become much harder to perform anomaly detection based on the reconstruction errors.

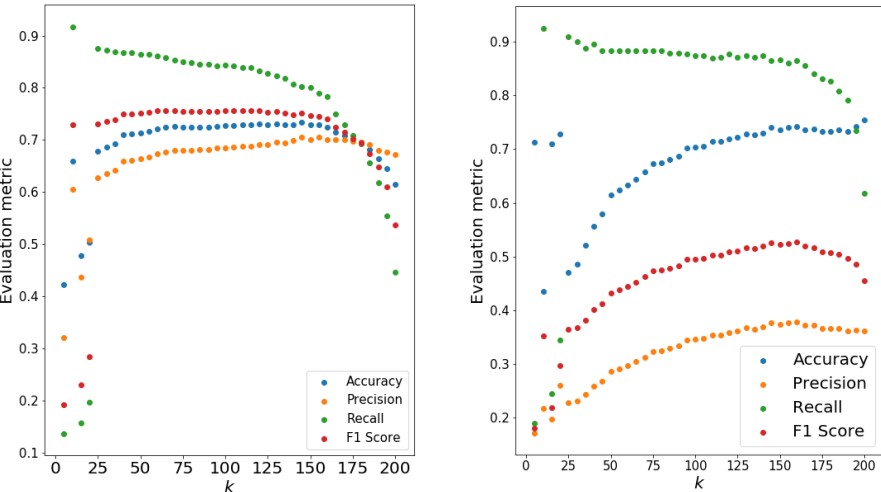

**Figure 4.** Performance metrics for the training (**left**) and test (**right**) sets with respect to the number of principal components $k$.

## 5. Model Evaluation: Main Results

We evaluated the performance of the identification and localization stages of our approach on the synthetic data using appropriate performance metrics. We demonstrated numerically the efficiency of the PCA NN over the baseline anomaly detection algorithms.

### 5.1. Contaminated Time Series Identification Step

For the feature extraction step, we considered a latent space dimension $k = 40$. The NN built to compute the anomaly scores and convert them into labels was calibrated on the training set. The result of this calibration is shown in Table 3.

**Table 3.** Performance evaluation of suggested model on synthetic data set for identification step.

| Data Set | Accuracy | Precision | Recall | $F_1$-Score |
|---|---|---|---|---|
| Train set | 90.97% | 97.36% | 84.21% | 90.31% |
| Test set | 88.58% | 61.26% | 85.27% | 71.30% |

Figure 5 shows two contaminated time series identified as such by the model. Figure 6 displays two examples of time series without anomalies accurately identified by the model. Figure 7 displays two time series misidentified by the model.

When an observation deviated significantly from the rest of the time series values, the model was able to recognize that the concerned time series contained an abnormal observation.

In Figure 6, with the uncontaminated time series we can see that even when there was a local upward trend in the time series values, the model was able to make the distinction between this market move and the occurrence of an anomaly.

The last set of time series displays some limits of the model. The top graphs in Figure 7 represent the situation where the model did not manage to identify the contaminated time series. One explanation could be that the size of the anomaly was not significantly large enough to be spotted by the model. Indeed, looking at the graph, the time series does not seem to contain any anomalies. In contrast, the bottom graphs in Figure 7 show a stock price path predicted to be contaminated, whereas it was not.

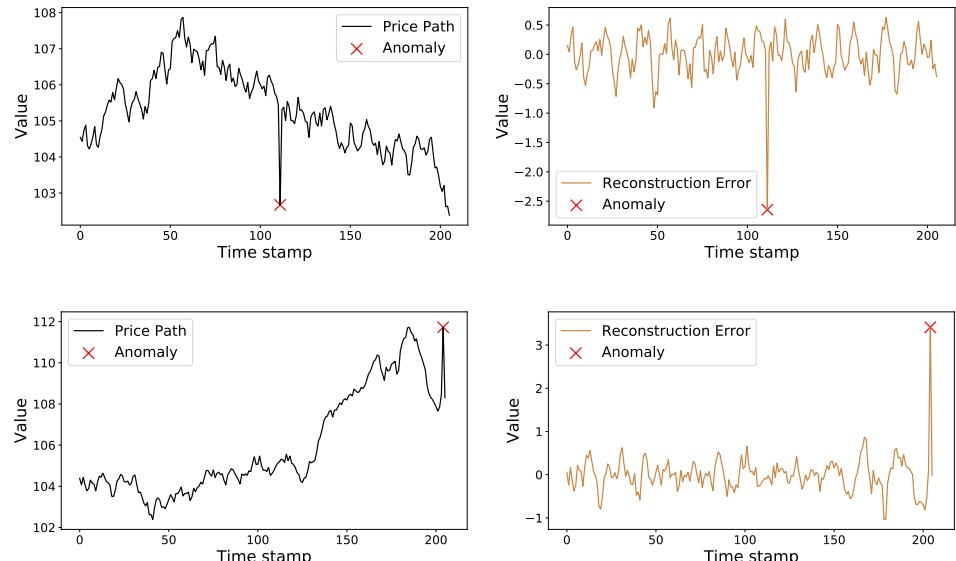

**Figure 5.** Two examples of contaminated time series accurately identified by the model. The stock path and the reconstruction errors are represented in black and brown. The red cross shows the anomaly localization.

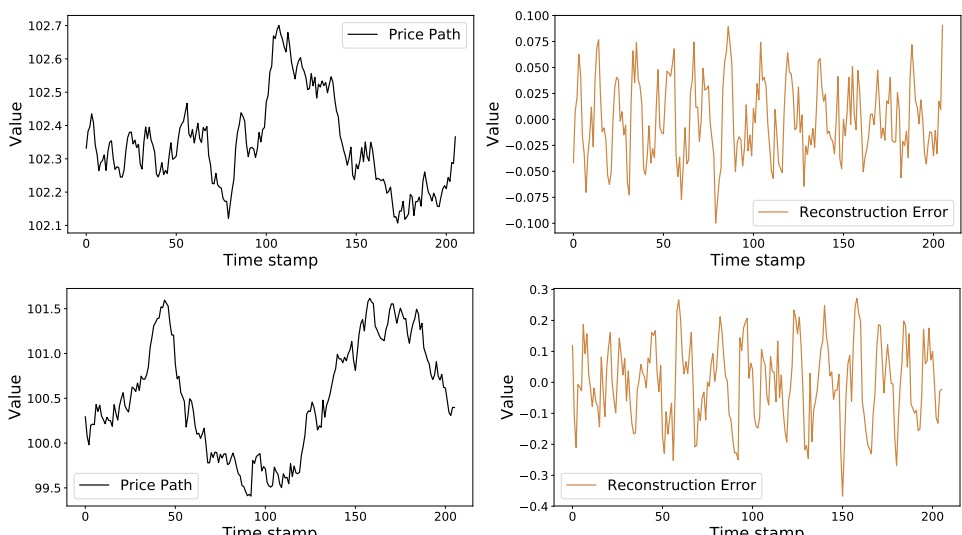

**Figure 6.** Two examples of uncontaminated time series accurately identified by the model. The stock path and the reconstruction errors are represented in black and brown.

We illustrated the robustness of the approach by assessing the model predictions on 100 distinct data sets. Table 4 shows the mean and standard deviations over the multiple runs.

**Table 4.** Mean (standard deviation) of performance metrics of the suggested model over multiple runs for the identification step.

| Data Set | Accuracy | Precision | Recall | $F_1$-Score |
|---|---|---|---|---|
| Train set | 79.02% (2.4%) | 78.69% (2.7%) | 79.74% (4.6%) | 79.13% (2.7%) |
| Test set | 77.79% (3.9%) | 41.87% (5.2%) | 80.16% (6.4%) | 54.82% (5.2%) |

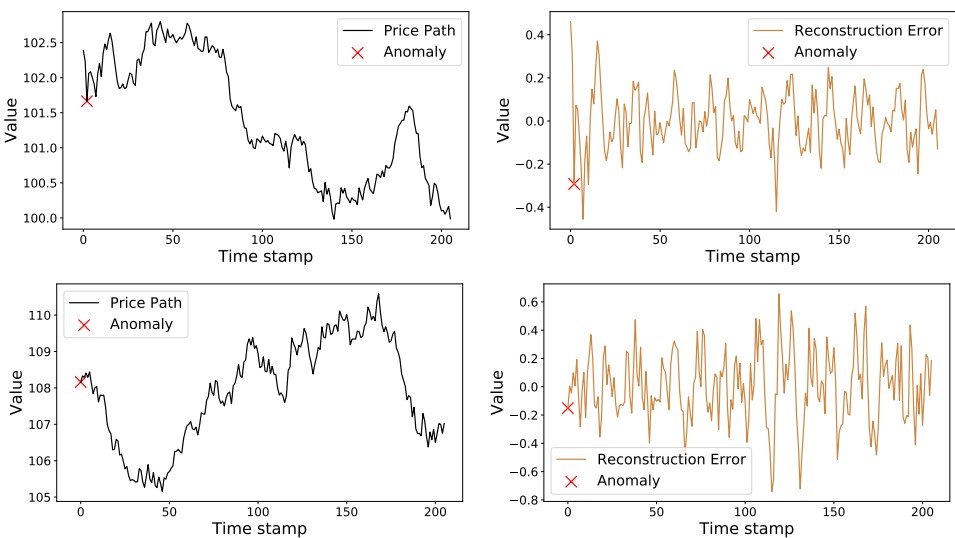

**Figure 7.** Two examples of time series misidentified by the model. The stock path and the reconstruction errors are represented in black and brown. The red cross shows the anomaly localization.

*5.2. Anomaly Localization Step*

Regarding the localization step, the dummy approach consisted of taking the argmax of the time series observations as the anomaly location. We distinguished between two cases, anomalies that were extrema and anomalies that were not. The quality of the

detection of our model was assessed for both types of anomalies. The results are displayed in Tables 5 and 6.

**Table 5.** Performance evaluation of the suggested model and (the dummy approach) on synthetic data set for the localization step of all types of anomalies.

| Data Set | Accuracy | Precision | Recall | $F_1$-Score |
|---|---|---|---|---|
| Train set | 89.65% (34.87%) | 89.81% (40.53%) | 89.65% (34.88%) | 89.68% ( 36.56%) |
| Test set | 94.39% (27.78%) | 94.49% (31.52%) | 94.39% (27.78%) | 94.38% (28.94%) |

**Table 6.** Performance evaluation of the suggested model and the dummy approach on synthetic data set for the localization step of non-extrema anomalies.

| Data Set | Accuracy | Precision | Recall | $F_1$-Score |
|---|---|---|---|---|
| Train set | 84.22% (0%) | 84.55% (0%) | 84.28% (0%) | 89.68% (0%) |
| Test set | 91.92% (0%) | 92.10% (0%) | 91.92% (0%) | 91.90% (0%) |

The numerical tests showed the necessity of the feature extraction step for the anomaly localization task, as applying the dummy approach alone was not enough when the anomaly was not an extreme value. Figure 8 represents a stock price time series with the true and predicted anomaly locations. In these examples, the locations were accurately predicted.

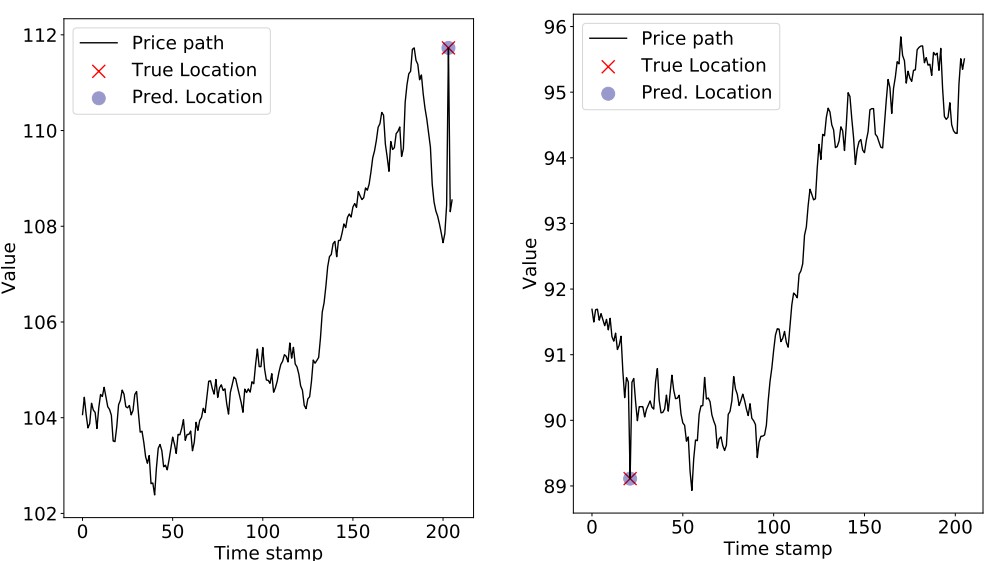

**Figure 8.** Anomaly localization prediction for two distinct time series.

To guarantee the robustness of the model on the anomaly location, we evaluated the model prediction on 100 data sets. Table 7 shows the mean and standard deviations for the localization step of all types of anomalies.

**Table 7.** Mean (standard deviation) of performance metrics of suggested model over multiple runs for localization step of all types of anomalies.

| Data Set | Accuracy | Precision | Recall | $F_1$-Score |
|---|---|---|---|---|
| Train set | 89.58% (4.3%) | 89.58% (4.3%) | 89.96% (4.1%) | 89.67% (4.3%) |
| Test set | 89.49% (4.8%) | 89.49% (4.8%) | 90.00% (4.5%) | 89.58% (4.7%) |

Table 8 displays the results of the localization of the non-extrema anomalies. These results show the importance of the feature extraction step. If we only considered the

maximal observed value of the time series to be the anomaly, we would not have been able to localize any non-extrema anomalies. Applying Algorithm 4 instead led to satisfactory anomaly detection rates.

**Table 8.** Mean (standard deviation) of performance metrics of suggested model over multiple runs for localization step of non-extreme anomalies.

| Data Set | Accuracy | Precision | Recall | $F_1$-Score |
|---|---|---|---|---|
| Train set | 85.23% (6.0%) | 85.23% (6.0%) | 85.87% (5.7%) | 85.36% (6.0%) |
| Test set | 85.17% (7.1%) | 85.17% (7.1%) | 86.04% (6.8%) | 85.30% (7.1%) |

*5.3. Numerical Results against Benchmark Models*

To assess the performance, the suggested approach was compared numerically with well-known machine learning algorithms for anomaly detection, that is, isolation forest (IF), local outlier factor (LOF), density-based clustering of applications with noise (DBSCAN), K-nearest neighbors (KNN), and support vector machine (SVM), all reviewed in Appendix B. In addition to these state-of-the-art techniques, we considered the recent anomaly detection technique proposed by Akyildirim et al. [14], referred to as sig-IF, which combines a feature extraction step through a path signature computation with IF.

Tables 9–11 summarize the performance on the training and test sets in Section 3 of each model for both the contaminated time series identification and the anomaly localization steps.

**Table 9.** Performance evaluation of unsupervised (**upper part**) and supervised (**lower part**) models for contaminated time series identification step.

| | Train | | Test | |
|---|---|---|---|---|
| **Model** | **Accuracy** | **$F_1$-Score** | **Accuracy** | **$F_1$-Score** |
| IF | 42.31% | 42.31% | 69.64% | 7.022% |
| LOF | 59.95% | 59.95% | 90.00% | 62.41% |
| DBSCAN | 50.00% | 66.67% | 16.65% | 28.53% |
| sig-IF | 49.48% | 49.23% | 72.32% | 15.19% |
| KNN | 94.68% | 94.39% | 64.82% | 26.69% |
| SVM | 81.96% | 82.33% | 44.39% | 27.44% |
| PCA NN | 90.97% | 90.31% | 88.58% | 71.30% |

**Table 10.** Execution time in seconds for identification of contaminated time series step.

| Algorithm | IF | LOF | DBSCAN | KNN | SVM | sig-IF | PCA NN |
|---|---|---|---|---|---|---|---|
| Exec. Time | 0.6915 | 0.2015 | 0.1275 | 1.725 | 4.572 | 2.776 | 0.003523 |

The following paragraphs provide similar conclusions, which are drawn from the results of the identification and localization steps (see Tables 9 and 11).

For the unsupervised learning methods, there was not a proper learning step and even though Tables 9 and 11 show the scores of the training and test sets, these scores should be seen as the ones obtained by testing the models on the independent data sets. For the IF, LOF, and DBSCAN models, the performance across the sets was not stable, as a significant difference could be observed between the scores on the training and test sets. The poor performance of the unsupervised learning algorithms could be explained by the difficulty in estimating the contamination rate for the IF and the LOF algorithms. For DBSCAN, the poor performance was rather due to the high dimensionality of the data. As for the sig-IF approach, its performance in this specific case was not as overwhelming as when used to detect pump-and-dump operations [14]. This difference in performance was driven by the fact that in our case, signatures were computed on the stock price path only since no

additional information was available to describe the price path we were analyzing. For the pump/dump detections, instead, the signatures were computed on a set of variables' paths, including the price path and additional variables path, which was helpful in the identification of the pumps/dump attempts.

Regarding the supervised approaches, although KNN outperformed the suggested approach on the training set, there was a non-negligible decrease in these scores on the test set. This may suggest an overfitting of the training data, which made KNN unable to generalize what it had learnt to the unseen samples. The same trend was seen in the scores with our approach; however, the loss was not as harsh as in the KNN case.

Hence, according to the results, our PCA-based approach seemed to be the most suitable method for the problem at stake of anomaly detection in time series. Its satisfactory performance in terms of accuracy and $F_1$-score, as well as its low computational costs for both steps (see Tables 10 and 12), make the approach stand out.

**Table 11.** Performance evaluation of unsupervised (upper part) and supervised (lower part) models for anomaly localization step. Results for DBSCAN, sig-IF, and SVM are not provided due to high computational costs.

| Model | Accuracy | $F_1$-Score | Accuracy | $F_1$-Score |
|---|---|---|---|---|
| IF | 89.79% | 2.296% | 70.94% | 1.611% |
| LOF | 99.51% | 0% | 2.066% | 0.9816% |
| DBSCAN | N/A | NA | NA | NA |
| sig-IF | N/A | N/A | N/A | N/A |
| KNN | 99.99% | 99.99% | 95.56% | 2.794% |
| SVM | NA | NA | NA | NA |
| PCA NN | 89.65% | 89.68% | 94.39% | 94.38% |

**Table 12.** Execution time in seconds for the anomaly localization step. Results for DBSCAN, sig-IF, and SVM are not provided due to high computational costs.

| Algorithm | IF | LOF | DBSCAN | KNN | SVM | sig-IF | PCA NN |
|---|---|---|---|---|---|---|---|
| Exec. Time | 14.50 | 2.287 | N/A | 14.19 | N/A | N/A | 0.002004 |

## 6. Model Evaluation: Additional Results

To take our approach a leap beyond classical anomaly detection algorithms, we evaluated the anomaly imputations suggested by the PCA NN approach. We also tested the robustness of the calibrated cutoff value and examined the sensitivity of the PCA NN performance in the amplitude of the anomaly.

### 6.1. Anomaly Imputations

To assess the imputation values suggested by our approach, we started with the time series simulated without anomalies, then we randomly selected a time stamp of the time series and added a noise to the corresponding value following the methodology described in Section 3.

The imputation value suggested by the PCA NN was the reconstructed observation. This imputation technique was compared to the naive methods of the imputation of missing values, namely backward fill (BF), which consists of replacing the anomaly with the previous value, and linear interpolation (LI). The choice of these imputation methods was motivated by their low computational costs. In fact, the PCA NN imputation came at no additional cost as stated before. Therefore, we only challenged its performance using methods with similar complexities. To assess the quality of the imputation value

suggested by each approach, we considered the following metrics. The imputation errors were computed for each time series $i$ as

$$\text{ImputationError}^i = \sqrt{\sum_j \frac{\left(S^i_{t_j} - \widetilde{S}^i_{t_j}\right)^2}{n^{anom}}},$$

where $\widetilde{S}^i$ refers to the $i$-th path price with imputed values. The error on the covariance matrix was computed as

$$\text{ErrCov} = \|\Sigma - \widetilde{\Sigma}\|_{\text{Frob}},$$

where $\| \cdot \|_{\text{Frob}}$ is the Frobenius norm, $\Sigma$ is the sample covariance matrix, and $\widetilde{\Sigma}$ is the co-variance matrix estimated for the data after the anomalies were replaced by their respective imputation values.

Each stock path was diffused and contaminated 100 times. The anomalies were then imputed following two baseline approaches (BF and LI). Figure 9 clearly shows that the imputation using the reconstructed values with PCA reduced the imputation error but not as much as the basic imputation techniques (LI and BF), as shown in Figure 10. Table 13 shows that the baseline imputation approaches achieved even lower errors in the estimation of the covariance matrix. Figure 11 shows a higher mean but a lower variance of the errors for the PCA-based imputation approach. In conclusion, it is recommended to replace the flagged anomalies with approaches such as backward fill or linear interpolation.

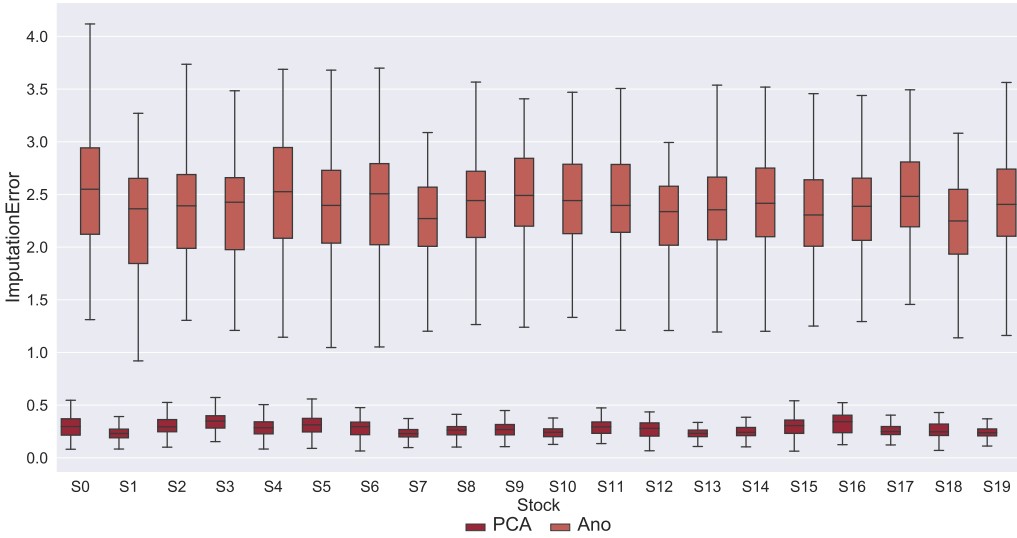

**Figure 9.** Box plot representation of the distribution of the errors on the time series imputation before the imputation of anomalies (Ano) and after replacing the anomalies with the reconstructed values suggested by the PCA NN (PCA).

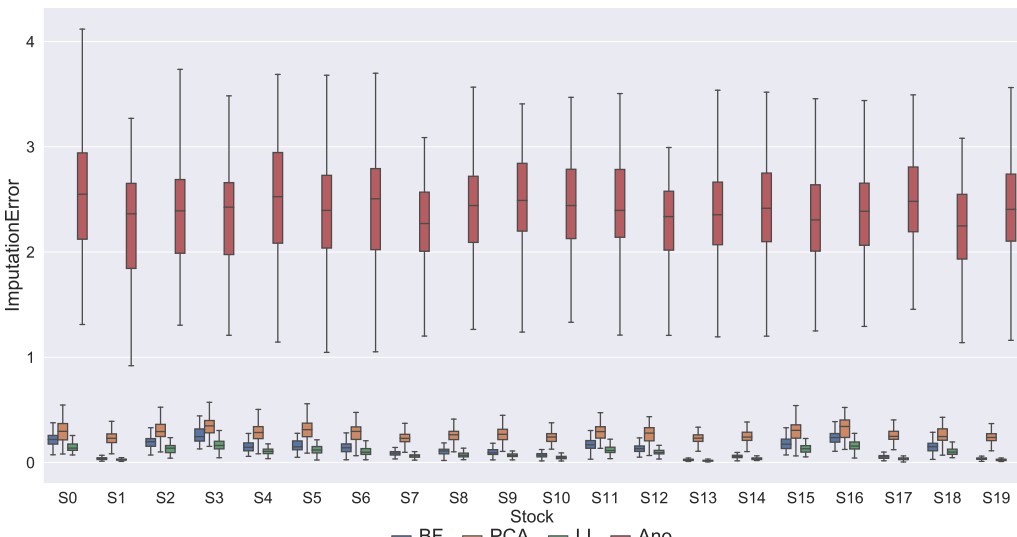

**Figure 10.** Box plot representation of the distribution of the errors on the time series imputation before the imputation of anomalies (Ano) and after replacing the anomalies with the reconstructed values suggested by PCA using linear interpolation (LI) and backward fill (BF).

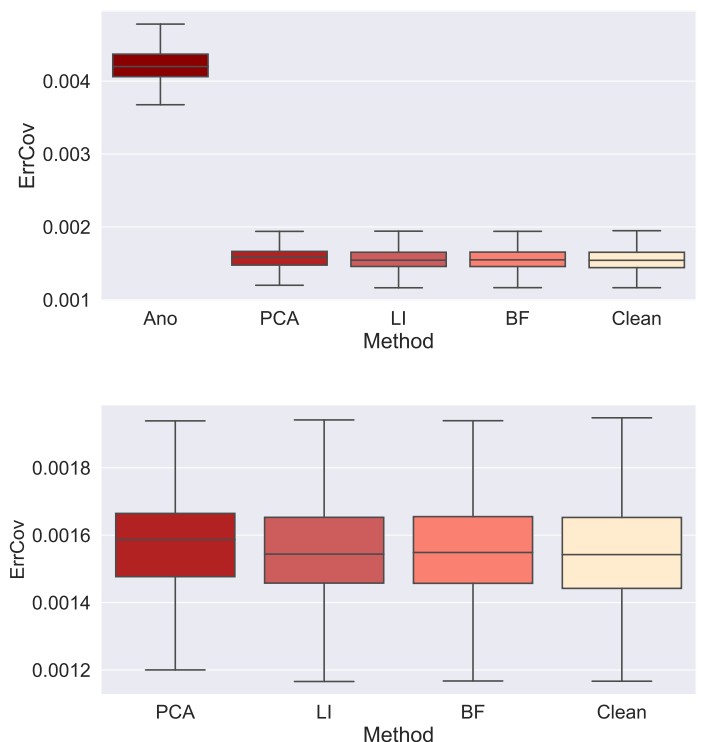

**Figure 11.** Box plot representation of the distribution of the errors on the covariance matrix (on several stock path samples). Ano, PCA NN, LI, BF, and Clean refer to the errors in the covariance when estimated for time series with anomalies; time series after anomaly imputation following three approaches, imputation by the reconstructed values suggested by PCA, linear interpolation (LI), backward fill (BF); and time series without anomalies (Clean). The Ano, PCA NN, LI, BF, and Clean errors in the covariance estimation are all represented (**upper**). For better visualization we remove the Ano errors (**below**).

**Table 13.** Mean and standard deviation errors on covariance matrix after imputation of anomalies with baseline methods.

| Method | Ano | PCA | LI | BF | Clean |
|---|---|---|---|---|---|
| Mean | 0.004208 | 0.001575 | 0.001554 | 0.001554 | 0.001552 |
| Standard deviation | 0.000272 | 0.000165 | 0.000170 | 0.000172 | 0.000173 |

A natural explanation for the relative inefficiency of the reconstruction value as the imputation value is that by construction, the reconstructed value integrates the abnormal value into its computation, which is not the case when using other imputation techniques. Therefore, even if the reconstructed value is closer to the true value than it is to the abnormal value, the spread between the imputation and the true value is still significant. Figures 12 and 13 illustrate the latter, with plots of the reconstructed and original price paths.

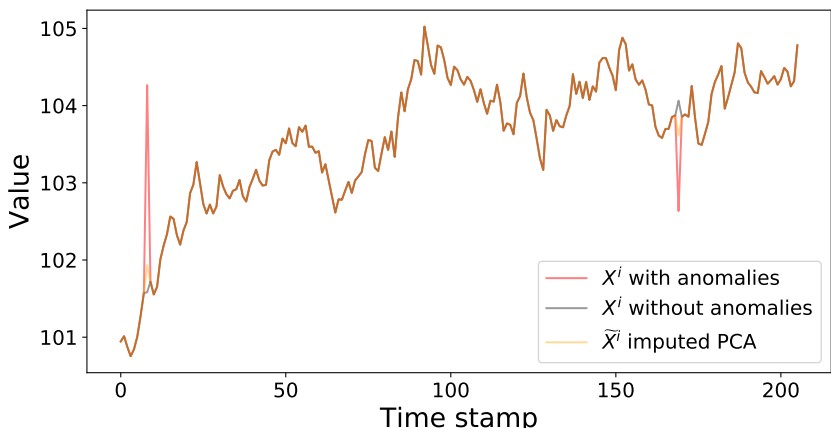

**Figure 12.** Original, contaminated, and reconstructed stock price paths.

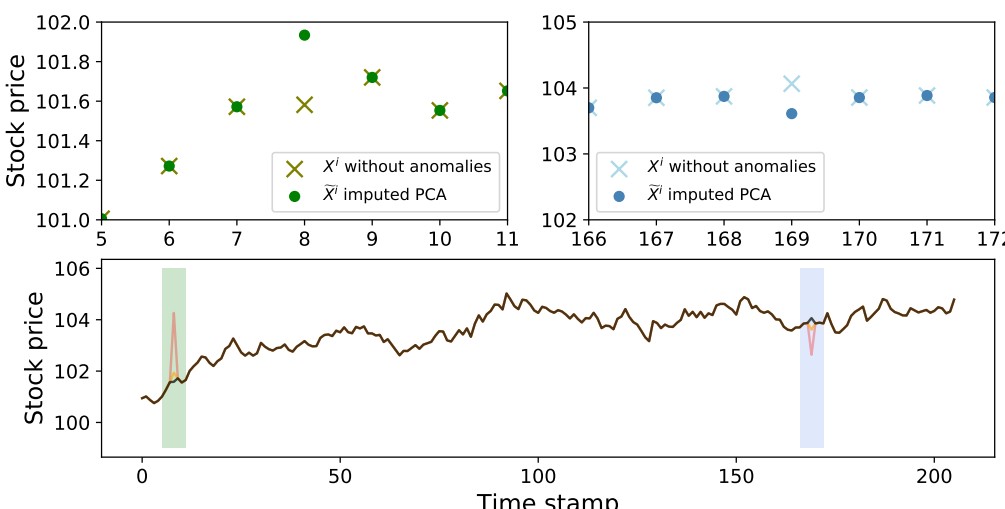

**Figure 13.** Original, contaminated, and imputed stock price paths in black, red, and orange. The two additional graphs represent the region around the anomalies, where the crosses and circles, respectively, show the true value of the stock and its value after imputation of the anomalies using the reconstructed value suggested by the PCA NN approach.

### 6.2. Cutoff Value Robustness

In most anomaly detection models, the cutoff value is a hand-set parameter. In our approach, the cutoff value is a model parameter and as such, it is calibrated through

the learning. Testing the robustness of the cutoff given by the model is therefore a must. The identification step is a unique step in the approach and concerns the robustness since it is the only step involving the cutoff calibration. The robustness was checked by shocking the suggested cutoff with different levels of noise and observing the impact of these shocks on the model's performance. We considered several shock amplitudes $\gamma \in \pm\{10^{-4}, 10^{-3}, 10^{-2}, 10^{-1}, 1, 2\}$. Table 14 reports the mean and standard deviations of the scores of the model. The cutoff calibrated on the synthetic training data sample was shocked. Scores for both the training and test sets were computed using the shocked cutoff values.

**Table 14.** Performance evaluation after shocking the calibrated cutoff values for the training set (**upper**) and the test set (**below**).

| $\gamma$ | Accuracy | Precision | Recall | $F_1$ |
|---|---|---|---|---|
| $10^{-4}$ | 0.9097 | 0.9736 | 0.8421 | 0.9031 |
| $-10^{-4}$ | 0.9097 | 0.9736 | 0.8421 | 0.9031 |
| $10^{-3}$ | 0.9097 | 0.9738 | 0.8419 | 0.9031 |
| $-10^{-3}$ | 0.9098 | 0.9737 | 0.8424 | 0.9033 |
| $10^{-2}$ | 0.9092 | 0.9747 | 0.8403 | 0.9025 |
| $-10^{-2}$ | 0.9098 | 0.9720 | 0.8439 | 0.9035 |
| $10^{-1}$ | 0.9042 | 0.9860 | 0.8201 | 0.8954 |
| $-10^{-1}$ | 0.9103 | 0.9563 | 0.8599 | 0.9056 |
| 0 | 0.9097 | 0.9736 | 0.8421 | 0.9031 |
| 1 | 0.7771 | 0.9994 | 0.5546 | 0.7133 |
| $-1$ | 0.5183 | 0.5093 | 0.9998 | 0.6749 |
| 2 | 0.6284 | 1.0000 | 0.2567 | 0.4086 |
| $-2$ | 0.5001 | 0.5000 | 1.0000 | 0.6667 |
| $\gamma$ | Accuracy | Precision | Recall | $F_1$ |
| $10^{-4}$ | 0.8858 | 0.6126 | 0.8527 | 0.7130 |
| $-10^{-4}$ | 0.8858 | 0.6126 | 0.8527 | 0.7130 |
| $10^{-3}$ | 0.8858 | 0.6126 | 0.8527 | 0.7130 |
| $-10^{-3}$ | 0.8850 | 0.6105 | 0.8527 | 0.7116 |
| $10^{-2}$ | 0.8877 | 0.6179 | 0.8527 | 0.7166 |
| $-10^{-2}$ | 0.8838 | 0.6074 | 0.8527 | 0.7095 |
| $10^{-1}$ | 0.8984 | 0.6502 | 0.8432 | 0.7342 |
| $-10^{-1}$ | 0.8672 | 0.5653 | 0.8741 | 0.6866 |
| 0 | 0.8858 | 0.6126 | 0.8527 | 0.7130 |
| 1 | 0.9391 | 0.9435 | 0.6746 | 0.7867 |
| $-1$ | 0.2660 | 0.1848 | 1.0000 | 0.3120 |
| 2 | 0.8949 | 0.9814 | 0.3753 | 0.5430 |
| $-2$ | 0.1700 | 0.1670 | 1.0000 | 0.2862 |

Excluding the extreme cases where the shock amplitude was $\pm\{1, 2\}$, one can see that the accuracy was barely impacted by the shocked cutoff values, for both the training and test sets. The interpretation of the remaining results is split into two and is applicable for both the training and test sets.

When negative shocks were applied, the model predicted more anomalies and fewer normal observations (compared to the predicted numbers with the calibrated cutoff value). Therefore, the model was able to identify the anomalies that were initially missed. Hence, applying negative shocks increased the recall. As for the precision, i.e., the rate of correctly identified contaminated time series, since on the left hand side of the cutoff value we were in the density region where contaminated time series represented the dominant class, the

new position of the cutoff value led to a misidentification of this type of observation. This entailed a deterioration in the precision. Opposite behaviours of the precision and recall were observed when positive shocks were applied since some abnormal time series were missed by the model.

Figure 14 shows the precision and recall when cutoff values other than the one we calibrated were used in our model on the training and test sets. These scores were also compared with the performance of the no-skill model (the no-skill model assigns a positive label to all observations). We noted that our approach edged out the no-skill model for both sets. The area under the curve (AUC) scores (AUC score ranged from 0 to 1, with 1 being the score associated with a perfect model) for the training and test sets were, respectively, 0.97 and 0.87. This shows that we performed better on the training set, which was expected, but the performance on the test set was just as satisfactory. These results reinforce our conclusions regarding the robustness of our approach. One can see that for the training set, the calibrated cutoff value represented the point where an equilibrium was reached between the precision and recall. The calibrated cutoff allowed for achieving high precision and recall scores simultaneously. We conclude that the cutoff given by the model was suitable for the training samples and also for the unseen samples.

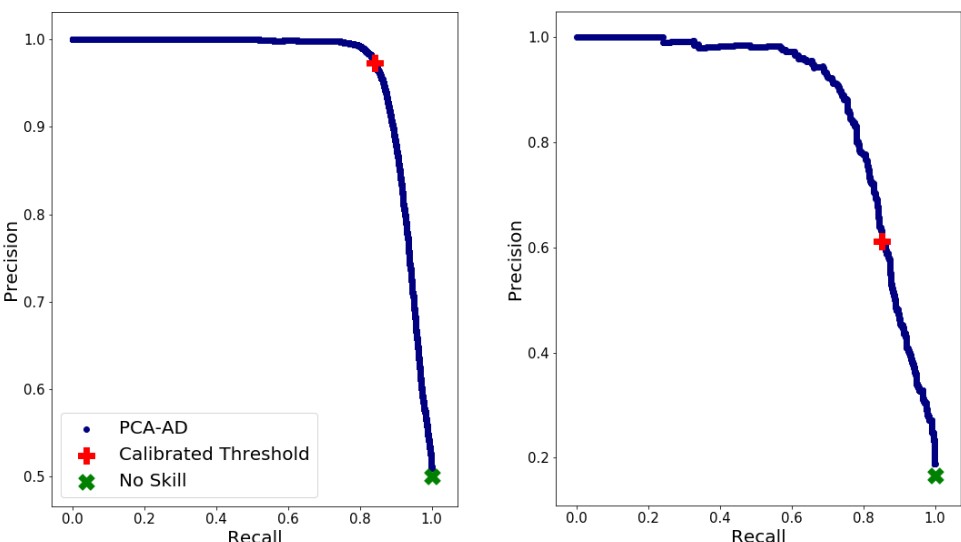

**Figure 14.** Precision–recall curve of the suggested model (in blue), the scores for the calibrated threshold (red plus), the no-skill model scores (green cross) for the training (**left**) and test sets (**right**).

To briefly sum up this section, although we mentioned that some shocks on the cutoff value induced higher scores, the improvement over the scores with the calibrated cutoff value was not significant (unless high-amplitude shocks are considered). The numerical tests and the precision–recall curves were consistent with the robustness of the cutoff value suggested by the approach.

*6.3. Sensitivity to Anomaly Amplitude*

Time series are manually contaminated as described in Section 3.2. The abnormal value $S_{t_j}^{a,i}$ of the *i*-th time series results from a shock of the initial value of the time series denoted by $S_{t_j}^i$ as stated in (9). We recall that the shock is represented by $\delta$, whereas its amplitude $|\delta|$ is uniformly drawn from $[0, \rho]$. Hence, $\rho$ is the parameter that ultimately controls the amplitude of the anomaly. Since $\rho$ is fixed by the user, it is interesting to investigate the sensitivity of the PCA NN approach's performance to the amplitude of the shock.

The PCA NN approach evaluated in Sections 5.1 and 5.2 was calibrated on time series that were contaminated with the shock amplitude drawn from $\mathcal{U}([0, \rho])$ with $\rho = 0.04$. For

the data sets on which the model was calibrated and then evaluated, the anomalies were grouped according to the shock amplitude they resulted from. For the identification step, we distinguished four groups of contaminated times. For instance, the first line of Table 15 defines the first group of contaminated time series for which the anomaly resulted from the shock amplitude $\in [0.309 \times 10^{-2}, 1.46 \times 10^2]$. For this first group with this specific range of amplitude, 77% of contaminated time series were identified during the PCA NN identification step.

**Table 15.** Detection ratios of the correctly identified contaminated time series in the test set, with the time series grouped according to their anomalies' shock amplitudes.

| Amplitude Range ($10^{-2}$) | Detection Ratio |
| :---: | :---: |
| [0.309, 1.46] | 0.77 |
| [1.46, 2.34] | 0.91 |
| [2.34, 2.92] | 0.96 |
| [2.92, 3.78] | 0.98 |

Similarly, for the localization step, we considered four groups of anomalies. The first line of Table 16 represents the anomalies with a shock amplitude $\in [0.309 \times 10^{-2}, 1.31 \times 10^{-2}]$. A total of 86% of the anomalies belonging to this first group were correctly localized by the PCA NN localization step.

**Table 16.** Detection ratios of the correctly localized anomalies on the test set, with the anomalies grouped according to their shock amplitudes.

| Amplitude Range ($10^{-2}$) | Detection Ratio |
| :---: | :---: |
| [0.309, 1.31] | 0.86 |
| [1.31, 2.29] | 1.00 |
| [2.29, 2.88] | 1.00 |
| [2.88, 3.78] | 1.00 |

The bounds defining each group were chosen to be the minimal value, 25%-quantile, 50%-quantile, 75%-quantile, and the maximum value over the shock amplitude. From the results in Tables 15 and 16, one can see that, except for the first group that represents the lowest shock amplitude, the detection ratios of the remaining groups are similar.

The calibrated PCA NN approach's performance was then tested on new data sets contaminated with distinct values of $\rho$. Figures 15 and 16 show that the $F_1$-scores are almost similar across the high values of $\rho$ for both the training and test sets. Moreover, Figure 16 clearly shows the similar and high performance of the localization stage on both data sets, regardless of the value of $\rho$.

To conclude the section on the sensitivity of the model's performance to the amplitudes of the anomaly shocks, although it is clear that a lower performance was observed in the identification and localization of smaller amplitude shock anomalies, the detection ratio was still satisfactory for this category of anomalies.

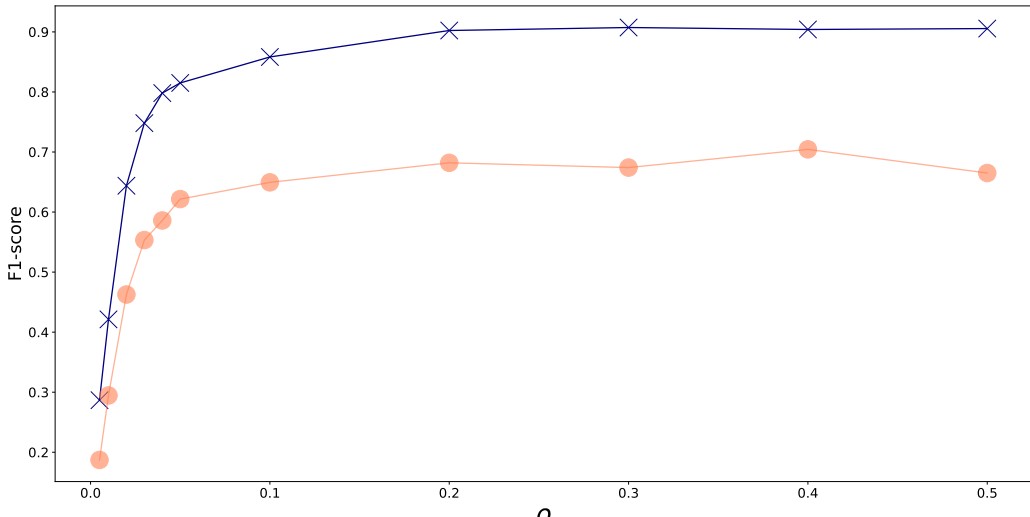

**Figure 15.** Mean $F_1$-score for identification step with respect to the various values of $\rho$ for the training and test sets, respectively, represented by the blue and pink curves.

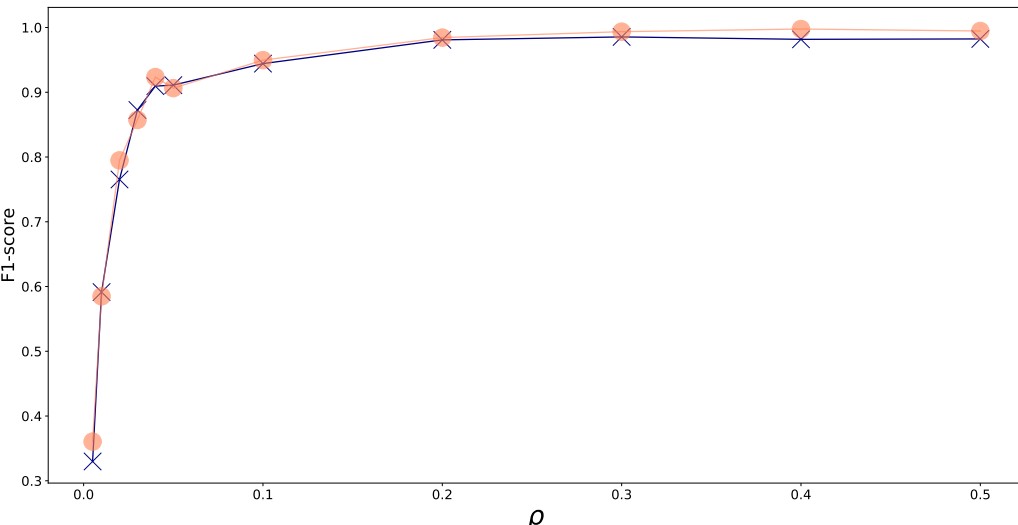

**Figure 16.** Mean $F_1$-score for localization step with respect to the various values of $\rho$ for the training and test sets, respectively, represented by the blue and pink curves.

## 7. Application to a Downstream Task: Value-at-Risk Computations

In this section, we illustrate the benefits of applying the PCA NN approach as a pre-processing step for value-at-risk computations.

Given a random variable $\varrho$ representing the loss in portfolio position over a time horizon $h$, its value-at-risk at the confidence level $\alpha \in (\frac{1}{2}, 1)$, $\text{VaR}_\alpha(\varrho)$, is defined by the quantile of level $\alpha$ of the loss distribution, i.e., $\mathbb{P}(\varrho \leq \text{VaR}_\alpha(\varrho)) = \alpha$ (assuming $\varrho$ is atomless for simplicity). Let $(S_t)_{t=t_1,\ldots,t_T}$ be a path price diffusion distributed according to the Black–Scholes model (7). The logarithmic returns to maturity are distributed according to the Gaussian distribution

$$\ln\left(\frac{S_{t+h}}{S_t}\right) \sim \mathcal{N}\left(\left(\mu - \frac{\sigma^2}{2}\right)h, \sigma^2 h\right), \tag{14}$$

with $\mu \in \mathbb{R}$ and $\sigma > 0$.

We assume the vector $\boldsymbol{R}$ of the log-returns of our stocks to be joint-normal,

$$\boldsymbol{R} = \left( R^1, R^2, \dots, R^i, \dots, R^N \right)^\top \sim \mathcal{N}_N(\mu_R, \Sigma_R),$$

where $R^i = \log\left( \frac{S^i_{t+h}}{S^i_t} \right) \sim \mathcal{N}\left( \mu_{i,R}, \sigma^2_{i,R} \right)$ and $\Sigma_R$ is the covariance matrix.

We consider a portfolio on $N$ stocks, whose return is given by $P = \mathcal{Q}^\top \boldsymbol{R}$, where $\mathcal{Q} \in \mathbb{R}^N$ defines the composition of the portfolio. Hence, $P \sim \mathcal{N}(\mu_P, \sigma^2_P)$, where $\mu_P = \mathcal{Q}^\top \mu_R$ and $\sigma^2_P = \mathcal{Q}^\top \Sigma_R \mathcal{Q}$. The value-at-risk for the time horizon $h$ at level $\alpha$ is

$$\text{VaR}_\alpha(P) = \mu_P + q_\alpha \sigma_P \,, \tag{15}$$

where $q_\alpha$ is the $\alpha$-quantile of a standard normal distribution and the parameters $\mu_R$ and $\Sigma_R$ are estimated from different types of time series to evaluate the impact of localizing and removing anomalies following our approach.

Under the adopted framework, the true $\text{VaR}_\alpha(P)$, $\text{VaR}^{theo}$, is thus known and can be computed using the diffusion parameters. An estimation $\widehat{\text{VaR}}_\alpha(P)$ can be obtained by replacing the parameters in (15) with their estimates $\widehat{\mu_P}$ and $\widehat{\sigma_P}$ computed from the time series with anomalies or after the imputation of anomalies. Then, the absolute errors and relative errors are computed as

$$\text{AbsoluteError}_{\text{VaR}} = \left| \text{VaR}_\alpha(P) - \widehat{\text{VaR}}_\alpha(P) \right|,$$

$$\text{RelativeError}_{\text{VaR}} = \frac{\left| \text{VaR}_\alpha(P) - \widehat{\text{VaR}}_\alpha(P) \right|}{\text{VaR}_\alpha(P)}$$

Table 17 summarizes the four VaR estimations we considered in the sequel.

**Table 17.** Notations of VaR estimations based on the time series from which the VaR parameters were estimated.

| VaR Estimation Name | $\widehat{\mu_R}, \widehat{\Sigma_R}$ Estimated on |
|---|---|
| $\text{VaR}^{clean}$ | Time series without anomalies |
| $\text{VaR}^{anom}$ | Times series with anomalies |
| $\text{VaR}^{loc,true}$ | Time series after anomaly imputation knowing their true localization |
| $\text{VaR}^{loc,pred}$ | Time series after anomaly imputation based on predicted localization |

To conduct this analysis, we generated new stock path samples that we assumed to be clean, fixing the diffusion parameters $\mu_R$ and $\Sigma_R$. We then added the anomalies following the procedure described in Section 3. We applied our model to localize the anomalies and replaced the localized anomalies using the backward fill (BF) approach (shown to be the more efficient imputation technique in Section 6.1). For each stock and each run of the simulation, we obtained four estimates of the distribution parameters of the associated log-returns.

Figure 17 and Table 18 summarize the distribution of the VaR estimates for $\alpha = 0.99$ and $h = 1(day)$ over several simulation runs. The box plots show the dispersion of the portfolio VaR estimates on several diffusions. The green square represents the mean of the VaR estimates. For the four first box plots, the means are approximately on the same level, which is confirmed by the results in Table 18. The anomalies present among the time series' observed values had a non-negligible impact on the distribution parameter estimation, which ultimately caused an incorrect estimation of the VaR. Thanks to the localization of the anomalies by the suggested model and their imputation as per Section 6.1, we were able to obtain a more accurate estimation of the VaR. The $\text{VaR}^{loc,true}$ and $\text{VaR}^{loc,pred}$ were quite similar, which shows that the model accurately localized the anomalies.

**Table 18.** Summary of VaR estimations for *P*.

| VaR | VaR$^{theo}$ | VaR$^{clean}$ | VaR$^{loc,true}$ | VaR$^{loc,pred}$ | VaR$^{anom}$ |
|---|---|---|---|---|---|
| Mean | 0.546851 | 0.546300 | 0.548392 | 0.548270 | 0.569015 |
| Standard Deviation | 0.0 | 0.010105 | 0.010739 | 0.010832 | 0.012268 |

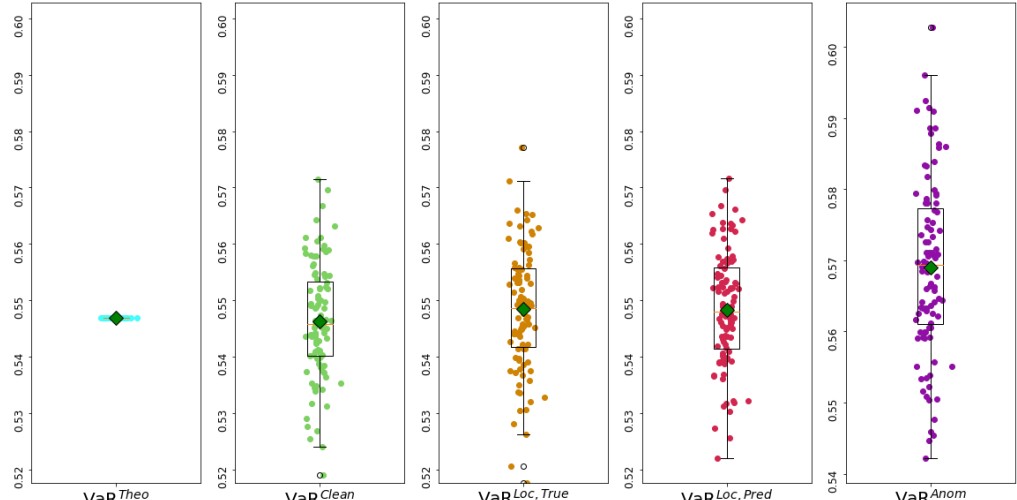

**Figure 17.** Box plot representation of the parametric VaR estimations for *P*. The green squares represent the mean of the VaR estimations.

We also evaluated the errors in the VaR estimations using the mean absolute error and the mean relative error, taking the VaR$^{theo}$ as our benchmark. As one can see in Table 19, even when the distribution parameters were estimated from the clean time series, the VaR computed with these parameters was not exactly the same as the one computed with the theoretical parameters. This can be explained by the historical size of the observed values used to estimate the parameters. This table shows that by removing anomalies we can reduce by a factor of two the errors in the VaR estimations.

**Table 19.** Mean absolute and relative errors in VaR estimations for *P*.

| VaR | VaR$^{clean}$ | VaR$^{loc,true}$ | VaR$^{loc,pred}$ | VaR$^{anom}$ |
|---|---|---|---|---|
| Absolute Error | 0.007995 | 0.008596 | 0.008622 | 0.02235 |
| Relative Error | 0.014620 | 0.015720 | 0.015767 | 0.04087 |

Additionally, we assessed the impact on VaR$^{theo}$, VaR$^{clean}$, VaR$^{anom}$, VaR$^{loc,true}$, and VaR$^{loc,pred}$ of increasing $n^{anom}$. To this end, we performed 50 simulation runs of stock paths for $n^{anom}$ and for each of those scenarios we estimated the VaR of the portfolio. We summarize the results in Figure 18, where each curve represents the mean VaR estimation with respect to $n^{anom}$, along with a representation of the uncertainty around each evaluated point through a confidence interval.

When we computed the VaR using the abnormal time series, we noticed that the difference between VaR$^{anom}$ and VaR$^{theo}$ increased with $n^{anom}$, which is natural to expect. However, when the time series were cleaned prior to the VaR estimations, the curves representing the VaR estimates were much closer to the ones representing VaR$^{theo}$ and VaR$^{clean}$, showing the undeniable improvement in the accuracy of the VaR estimations over the estimations based on abnormal time series. Furthermore, the VaR estimations after the imputation following the model prediction or knowing the true localization of the anomalies seemed to be quite similar for low $n^{anom}$, whereas some discrepancies between the two became more significant as $n^{anom}$ increased. A natural explanation could be that

when the number of anomalies increased and the model suggested incorrect anomaly localizations, normal values were replaced, whereas true anomalies remained among the observed values, which wrongly impacts the VaR estimations. However, the results shown in Tables 20 and 21 indicate that the anomaly localizations suggested by the model were overall correct and allowed for removing most of the anomalies, as the relative error of $VaR^{loc,pred}$, regardless of $n^{anom}$, was always lower than the relative error of $VaR^{anom}$ (e.g., a relative error of 0.0248 for $VaR^{loc,pred}$ against 0.0658 for $VaR^{anom}$, when there were 76 anomalies among the 1500 observed values of the time series).

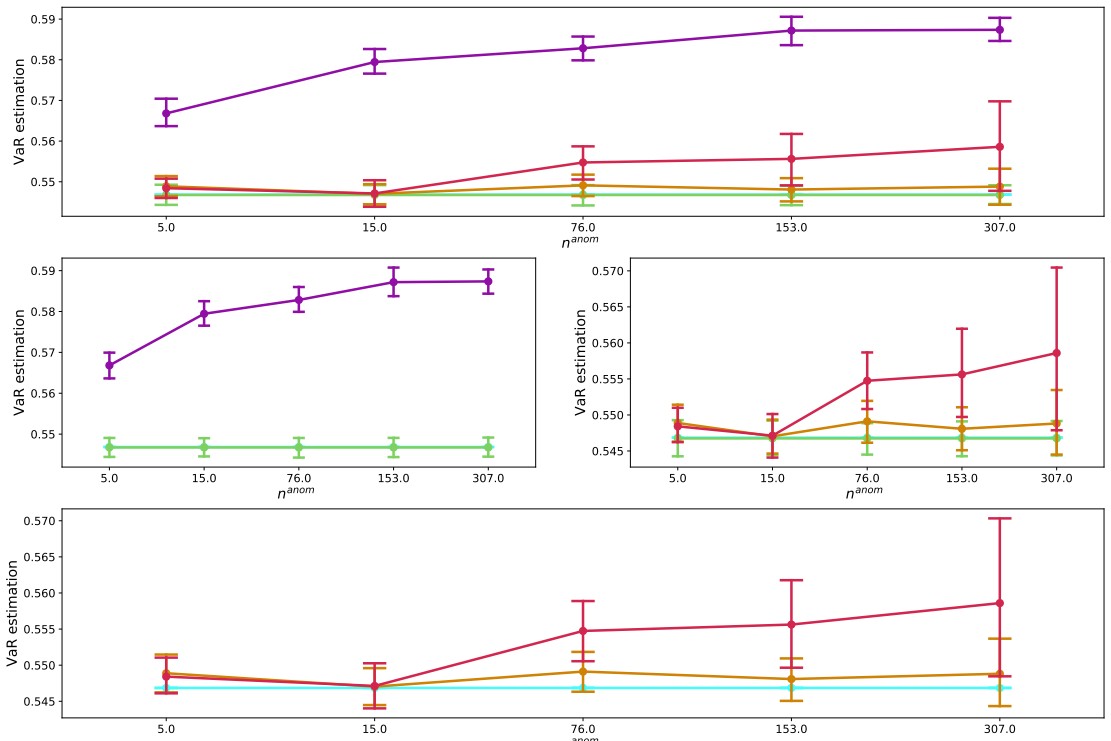

**Figure 18.** VaR estimations based on parameter estimations from time series with and without anomalies (purple and light-blue curves), time series after localization and imputation of anomalies with the suggested approach (red curve), and time series imputed knowing the true localization of the anomalies (brown curve) with respect to $n^{anom}$.

**Table 20.** Mean relative error of parametric VaR estimations with respect to $n^{anom}$ for parameters estimated from clean time series, time series with anomalies, imputed time series following predicted location ($VaR^{loc,pred}$), and true anomaly localization ($VaR^{loc,true}$).

| $n^{anom}$ | $VaR^{clean}$ | $VaR^{loc,true}$ | $VaR^{loc,pred}$ | $VaR^{anom}$ |
|---|---|---|---|---|
| 5 | 0.012194 | 0.013796 | 0.013341 | 0.037392 |
| 15 | 0.012194 | 0.012623 | 0.016676 | 0.059604 |
| 76 | 0.012194 | 0.014831 | 0.024807 | 0.065791 |
| 153 | 0.012194 | 0.014644 | 0.034361 | 0.073750 |
| 307 | 0.012194 | 0.020537 | 0.052244 | 0.074091 |

**Table 21.** Standard deviation of relative errors of parametric VaR estimations with respect to $n^{anom}$ for parameters estimated from clean time series, time series with anomalies, imputed time series following predicted location (VaR$^{loc,pred}$), and true anomaly localization (VaR$^{loc,true}$).

| $n^{anom}$ | VaR$^{clean}$ | VaR$^{loc,true}$ | VaR$^{loc,pred}$ | VaR$^{anom}$ |
|---|---|---|---|---|
| 5 | 0.010356 | 0.011511 | 0.009678 | 0.020674 |
| 15 | 0.010356 | 0.010514 | 0.012723 | 0.020172 |
| 76 | 0.010356 | 0.011842 | 0.018696 | 0.019930 |
| 153 | 0.010356 | 0.013683 | 0.026117 | 0.024243 |
| 307 | 0.010356 | 0.022526 | 0.057903 | 0.019207 |

## 8. Numerical Results Using Real Data

We considered a labelled real data set including stock prices, bond yields, CDS spreads, FX rates, and volatilities. These data were collected from a financial data provider for the period between 2018 and 2020. These data sets were labelled by experts. They are provided on https://github.com/MadharNisrine/PCANN (accessed on 12 October 2022) in the form of 132,000 time series (after augmentation). The training set is balanced, whereas the test set is imbalanced with 20% of contaminated time series.

To ensure the independence of the training and test data sets, we calibrated the PCA NN approach considering the time series of 2018 and 2019, whereas the 2020 time series were used in the model evaluation. The performance evaluation of the PCA NN approach after its calibration on the real data sets is shown in Table 22. As is visible in the upper part, the PCA NN performed quite well for the identification step. Once the contaminated time series were identified, the model was able to localize the anomaly with high accuracy, as reflected in the scores of the test set in the lower part.

**Table 22.** Performance of the PCA NN identification step (**upper part**) and localization step (**lower part**) on the real data set.

| Data Set | Accuracy | Precision | Recall | $F_1$-Score |
|---|---|---|---|---|
| Train set | 92.88% | 99.17% | 86.49% | 92.40% |
| Test set | 88.15% | 72.45% | 46.10% | 56.35% |
| **Data Set** | **Accuracy** | **Precision** | **Recall** | **$F_1$-Score** |
| Train set | 99.48% | 99.49% | 99.48% | 99.48% |
| Test set | 96.05% | 96.21% | 96.05% | 95.92% |

*Pca NN against State of the Art Models on Real Data*

Even if the $F_1$-score in the identification step was not that high, the results show that the related performance was better than those of alternative state-of-the-art approaches.

Tables 23 and 24 compare the performance of the PCA NN approach against the state-of-the-art models in Appendix B on our real data set. The PCA NN outperformed the benchmark models in both steps, with overwhelming results for the anomaly localization step.

**Table 23.** Performance evaluation of unsupervised (**upper part**) and supervised (**lower part**) models in the contaminated time series identification step.

| Model | Accuracy | Precision | Recall | $F_1$-Score |
|:---:|:---:|:---:|:---:|:---:|
| IF | 73.17% | 18.12% | 17.53% | 17.82% |
| LOF | 82.33% | 44.57% | 26.62% | 33.33% |
| DBSCAN | 22.74% | 15.71% | 83.77% | 26.46% |
| sig-IF | 73.34% | 18.15% | 17.47% | 17.80% |
| KNN | 70.26% | 23.48% | 35.06% | 28.13% |
| SVM | 50.11% | 24.46% | 96.10% | 39.00% |
| PCA NN | 88.15% | 72.45% | 46.10% | 56.35% |

**Table 24.** Performance evaluation of unsupervised (**upper part**) and supervised (**lower part**) models in the contaminated time series localization step. Results for DBSCAN, sig-IF, and SVM are not provided due to high computational costs.

| Model | Accuracy | Precision | Recall | $F_1$-Score |
|:---:|:---:|:---:|:---:|:---:|
| IF | 78.50% | 2.173% | 98.35% | 4.252% |
| LOF | 87.04% | 0.3713% | 9.616% | 0.7151% |
| DBSCAN | N/A | N/A | N/A | N/A |
| sig-IF | N/A | N/A | N/A | N/A |
| KNN | 99.85% | 78.38% | 95.59% | 86.13% |
| SVM | N/A | N/A | N/A | N/A |
| PCA NN | 96.05% | 96.21% | 96.05% | 95.92% |

## 9. Conclusions

We propose a two-step approach for detecting anomalies in a panel of time series that can reflect a wide variety of market risk factors. The first step aims to identify the contaminated time series, i.e., time series with anomalies. The second step focuses on the localization of the anomalies among the observed values of the identified contaminated time series. As a pre-processing step, our methodology integrates the extraction of features from the time series with PCA. This part of the method proved to be essential, as it provides the models with inputs on which the distinction between abnormal/contaminated and normal instances is based while also ensuring the stationarity of the model's (time series) inputs. Another key point of the approach is the calibration of the cutoff value, which is the key parameter in the identification of contaminated time series by means of a feedforward neural network with a customized loss function. The proposed approach suggests an imputation value; however, this value is strongly influenced by the abnormal value. Therefore, basic imputation approaches with similar complexities are preferred. Our numerical experiments not only show that our approach outperforms the baseline anomaly detection models but also the real benefits that could be gained from applying it as a preliminary data cleaning step prior to VaR computations. Future research could focus on the replacement of PCA by partial least squares (PLS) or deep PLS [15] for endogenizing the feature extraction stage. Regarding downstream tasks, our approach might be of special interest for reverse stress tests [16].

**Author Contributions:** Conceptualization, S.C., N.L., N.M. and M.T.; methodology, S.C., N.L., N.M. and M.T.; software N.M.; validation, S.C., N.L., N.M. and M.T.; formal analysis, S.C., N.L., N.M. and M.T.; investigation, S.C., N.L., N.M. and M.T.; resources, S.C., N.L., N.M. and M.T.; data curation N.L. and N.M.; writing—original draft preparation, N.M.; writing—review and editing, S.C., N.L., N.M. and M.T.; visualization, S.C., N.L., N.M. and M.T.; supervision, S.C., N.L. and M.T.; project administration, S.C., N.L. and M.T.; funding acquisition, S.C., N.L. All authors have read and agreed to the published version of the manuscript.

**Funding:** The research of N. Madhar is funded by a CIFRE grant from Natixis. The research of S. Crépey has benefited from the support of the Chair "Capital Markets Tomorrow: Modeling and Computational Issues" under the aegis of the Institut Europlace de Finance, a joint initiative of Laboratoire de Probabilités, Statistique et Modélisation (LPSM)/Université Paris Cité and Crédit Agricole CIB.

**Data Availability Statement:** Data supporting reported results can be found on https://github.com /MadharNisrine/PCANN, (accessed on 12 October 2022).

**Conflicts of Interest:** The authors declare no conflict of interest. The funders had no role in the design of the study; in the collection, analyses, or interpretation of data; in the writing of the manuscript, or in the decision to publish the results.

## Appendix A. Literature Review

We start with a review of the anomaly detection literature (see also Appendix B for a more technical presentation of some of the below-mentioned algorithms).

### *Appendix A.1. Baseline Algorithms*

Anomaly detection aims to find an *"observation that deviates so much from other observations as to arouse suspicion that it was generated by a different mechanism"* [2]. The baseline anomaly detection algorithms described in [17] struggle to identify anomalies in time series, mainly because their assumptions are invalidated. If we consider models built for spatial data, a major assumption of these models is that observations are independent, whereas, for time series, high dependency exists between different time stamps. Clustering-based approaches, such as the density-based spatial clustering with noise (DBSCAN) method, are particularly impacted by this aspect; if an anomaly occurs at a given time stamp and is followed by incorrect values, clustering-based approaches consider that the observations of the time series belong to two different clusters and thus fail to identify the anomaly. Another limitation when considering these types of techniques is the choice of the similarity metric used for the data clustering. This task, although a crucial pillar of these approaches, is not trivial and becomes very challenging for high-dimensional problems.

### *Appendix A.2. Statistical Approach to Anomaly Detection*

The statistical techniques for anomaly detection can be split into two families: statistical tests and predictive models. Both suffer from the curse of dimensionality and model/data mismatch. When anomaly detection relies on hypothesis tests, it usually tests whether the observations are drawn from a known distribution [18], supposing that the user knows the probability distribution of the normal observations. Such a parametric framework narrows down the scope of applicability of hypothesis tests, as the data do not always coincide with the assumed distribution. Moreover, the tests provided in the literature are not suitable in multivariate settings [19]. The statistical techniques that rely on fitting a predictive model to each time series also require strong assumptions on the data. Predictive models are usually autoregressive (AR), moving average (MA), or ARMA models. Anomalies are then detected relative to the forecasts suggested by the model [17]. In these parametric approaches, some parameters have to be specified again, starting with the order of the models. Selecting the optimal model parameters with respect to an information criterion is not always possible. Additionally, these models assume that the time series are homogeneous, i.e., drawn from the same distribution [20]. This is not always satisfied in the financial risk management case where several types of market risk factors are treated simultaneously.

### *Appendix A.3. Score-Based Anomaly Detection Models*

Additional anomaly detection challenges are of general concern. Most of anomaly detection algorithms are score-based in the sense that these approaches return an anomaly score reflecting the extent to which the observation is considered abnormal by the model. In order to decide whether an observation is abnormal or not, a cutoff value of the score

has to be selected. Empirical approaches are often used, consisting of setting the cutoff value as the quantile or elbow point of the distribution of the anomaly score. However, the selected cutoff value according to such methods remains arbitrary. Ref. [21] proposed to rely on the cost of misclassification based on a weighted classification accuracy. Approaches that calculate the "optimal" weights are described in [22] but they involve a heuristic grid search technique. Another alternative is to determine the cutoff value by cross validation of the training data [23]. Finally, some methods do not select any cutoff value but are based instead on a contamination rate. However, fixing a cutoff value or deciding on a contamination rate is not so different.

### *Appendix A.4. Scarcity of Anomalies and Data Augmentation*

The scarcity of anomalies within the data sets is another typical problem in anomaly detection. Anomalies are, by definition, rare events; therefore, they are under-represented in the data set used to fit the models. This under-representation does not help with the design of a reliable model able to identify anomalies. Classical methods used to overcome this issue consider data augmentation. These techniques aim to produce new synthetic samples that will ultimately enhance model performance, leveraging a better representation of the feature space. As reported in [24], time series can be augmented using a simple transformation of time domain [25], frequency domain [26], or more advanced generative approaches involving deep learning techniques such as recurrent generative adversarial networks [27]. In practice, the use of generative models for the data augmentation of time series with anomalies presents two limitations. First, training such models requires a large number of samples to guarantee satisfactory performance. Although restricted Boltzman machines do not exhibit this problem, Ref. [28] showed that they fail at fitting multivariate complex distributions with nonlinear dependence structures. A more fundamental limitation affects the very idea behind the generative models. Such networks are trained to learn a given distribution. However, by definition, anomalies are different to each other and therefore there is not a distribution that characterizes them.

### *Appendix A.5. Supervised vs. Unsupervised Learning*

Since anomalies are the realizations of atypical events for which the distribution is unknown, it seems quite natural to use unsupervised algorithms. However, these approaches are deemed more suitable for learning complex patterns and are task specific. Moreover, Ref. [29] pointed out that they often do not present a high prediction performance, in particular in high-dimensional settings [30]. Indeed, the performance of unsupervised shallow anomaly detection algorithms depends on a feature engineering step. Ref. [14] used signatures to extract features, which then fed algorithms such as isolation forest. This combination of techniques was shown to overperform the benchmark approaches. However, the designed model was task specific (detection of pump-and-dump attacks) and the feature extraction step was only efficient when at least one explanatory variable was considered in the analysis. Moreover, the unavailability of labelled data made the model building and evaluation even more complex. As for supervised methods, the only limitation on which the literature tends to agree is their incapacity to generalize the learned patterns to new samples, which is the consequence of the misrepresentation of anomalies among the training samples [31]. However, the scarcity of labelled data can be sidestepped through the use of data augmentation techniques on the fraction of available labelled data. For these reasons, the supervised learning framework is preferred even when only a small set of labelled data is available.

### *Appendix A.6. Anomaly Detection in Time Series*

Usually, anomaly detection models in time series have two main components. The first component aims to extract a parsimonious yet expressive representation of the time series. Several approaches are suggested in the literature to deal with such feature extraction. Recently, deep neural networks have been shown to suffer from overparametrization

and often be computationally expensive [14,32]. Path signatures are also computationally demanding. This could perhaps be alleviated by random signatures [33]. However, the information extracted with signatures is of most interest when the considered paths are characterized by several variables. The resulting representation is then transformed into a (typically continuous) anomaly score, which is, in turn, converted into a binary label [34].

*Appendix A.7. PCA and Anomaly Detection*

In the literature, PCA is usually used in anomaly detection for its dimension reduction properties. Anomalies are identified on the latent space, either by applying some anomaly detection algorithm or by assuming a given distribution on the principal component and identifying the anomalies relative to a quantile [35]. Assuming that a normal subspace representation of the data set can be constructed with the first $k$-components [36], anomaly detection is achieved by looking at the observations that cannot be expressed in terms of the first $k$-components [37]. Note that the particular power-fullness of an auto-encoder, a nonlinear PCA, as a data compressor is not desirable herein since auto-encoders compress all patterns including abnormal ones. Although [38] claimed that the PCA-based models are stable with respect to their parameters, such as the number of principal components $k$ spanning the subspace or the cutoff level, Ref. [36] found instead that PCA-based anomaly detection is sensitive to these parameters and to the amplitude of the anomalies. Indeed, the latter may undermine the construction of the normal subspace representation, in turn leading to the misidentification of anomalies. In light of this, we took extra care regarding these aspects, and appropriate tests were conducted to show that the proposed approach was not subject to these issues. In the end, with a relatively low number of features given by PCA, we were able to accurately describe the dynamics of the market risk factors represented by times series. The reason behind this is the high correlation structure displayed by the market risk factors.

## Appendix B. State-of-the-Art Anomaly Detection Models

*Appendix B.1. Density-Based Models*

### Appendix B.1.1. Density-Based Spatial Clustering with Noise (DBSCAN)

DBSCAN [39,40] is an unsupervised clustering methodology that groups together comparable observations based on a similarity metric. Clusters are high-density regions and are defined by the $\varepsilon$-neighbourhood of observations and by *MinPts*, the minimum number of points required to be in a radius of $\varepsilon$ from an observation to form a dense region. An anomaly is any observation that neither has no *MinPts* in its $\varepsilon$-neighbourhood nor appears in the $\varepsilon$-neighbourhood of other observations.

### Appendix B.1.2. K-Nearest Neighbours (KNN)

Usually used for classification purposes, KNN [41] is a supervised algorithm that can also be used for anomaly detection. For each observation, it selects its closest $K$ observations, generally in terms of distance but other similarity metrics can be considered. The anomaly score of an observation is computed as the function of its distances to the $K$-nearest neighbours, which is the weighted average of the distances. The points with the highest anomaly scores are considered anomalies.

### Appendix B.1.3. Support Vector Machines (SVM)

The ultimate aim of SVM [42] is to define a hyperplan that separates the data. The specificity of this hyperplan is that it maximizes the distances to the set of features representing each class. When the data are not linearly separable, a map $\phi$ is applied to the initial feature vector so the data become linearly separable in the new space in which they were projected.

*Appendix B.2. Depth-Based Models*

Appendix B.2.1. Isolation Forest (IF)

The IF method [43] applies a depth approach to detecting anomalies. The algorithm is based on the idea that anomalies are easier to isolate and thus will be isolated closer to the root, whereas normal observations are isolated much further from the root. The algorithm uses random decision trees to separate the observations. The anomaly score is calculated as the path length to isolate the observation. As it is defined in the algorithm, the anomaly score will be closer to 1 for anomalies and $\ll 1$ for normal observations. This allows for ranking the observations from the most abnormal observation to the most regular one. However, the choice of an explicit cutoff between these two types of instances is not obvious.

Appendix B.2.2. Local Outlier Factor (LOF)

The LOF algorithm [44,45] tries to assess the isolation of one observation relative to the rest of the data set before flagging it as an anomaly. This model relies on the concept of local density. The anomaly score of an observation $x$ is its local outlier factor, which quantifies how dense the location area of $x$ is compared to that of its neighbours. Hence, for each observation, $\text{LOF}_k(x) \approx 1$ means that the density of observations around $x$ is similar to that of its neighbours; therefore, $x$ could not be considered an isolated observation. $\text{LOF}_k(x) \gg 1$, instead, shows that the density of $x$ is lower compared to its neighbours; hence, $x$ should be flagged as an anomaly.

**Appendix C. The Data Stationarity Issue**

Assume that a process $\left(\varepsilon_t^i\right)$ satisfies the following representation:

$$\Delta \varepsilon_t^i = \gamma \varepsilon_{t-1}^i + \theta_1 \Delta \varepsilon_{t-1}^i + \ldots + \theta_{p-1} \Delta \varepsilon_{t-p+1}^i + z_t,$$

where $p$ is the lag order, $\Delta$ is the difference operator, i.e., $\Delta \varepsilon_t = \varepsilon_t - \varepsilon_{t-1}$, and $z_t$ is a white noise. The stationarity of $\left(\varepsilon_t^i\right)$ is shown using the augmented Dickey–Fuller test [46], namely the process $\varepsilon^i$ is stationary if there is a unit root, i.e., $\gamma = 0$. Therefore, for each $\varepsilon^i$, the test is carried out under the null hypothesis $\mathcal{H}_0 : \gamma = 0$ against $\mathcal{H}_1 : \gamma < 0$.

**Table A1.** Descriptive statistics for $p$-values for training and test set time series $\varepsilon$.

| Set | Mean | Standard Dev. | Min | 25% | 50% | 75% | Max |
|---|---|---|---|---|---|---|---|
| Train | $1.26 \times 10^{-14}$ | $1.74 \times 10^{-13}$ | $6.63 \times 10^{-24}$ | $3.46 \times 10^{-20}$ | $1.92 \times 10^{-18}$ | $3.74 \times 10^{-17}$ | $3.88 \times 10^{-12}$ |
| Test | $1.51 \times 10^{-14}$ | $4.83 \times 10^{-13}$ | $6.20 \times 10^{-30}$ | $4.23 \times 10^{-19}$ | $8.45 \times 10^{-18}$ | $1.65 \times 10^{-16}$ | $4.35 \times 10^{-11}$ |

If the $p$-values are lower than the significance level, then the null hypothesis is rejected for all the reconstruction errors and we conclude that the time series we are working with does not suffer from the non-stationarity issue.

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
