# Peer review of "Anomaly Detection in Financial Time Series by Principal Component Analysis and Neural Networks†"

_algorithms, doi:10.3390/a15100385_

Round 1

Reviewer 1 Report

The authors propose one of the methods for finding anomalies in time series. The time series is divided into intervals and, based on the neural network, anomalies are searched for in each of the parts.

The advantages of the work are that the method for detecting anomalies in time series is described in detail and has a software implementation. All the results can be reproduced. The method showed a rather high sensitivity to anomalies. 88% of anomalies are detected as a result of applying the method.

The disadvantages of work are:

1. This method works after the fact and the ergodicity of the time series is not taken into account.

2. The model is built and tested on artificial data. Outliers are also generated on artificial data. If this model is intended to manage risks, it is necessary to test it on real exchange time series.

3. To determine the characteristics of anomalies, moments of higher orders are not considered. Only the eigenvalue is evaluated.

4. There is no comparative analysis with other methods. May be the classical outlier detection methods (quartile method and higher order method) work better. There is no explanation why a neural network is used to detect anomalies and whether there are more efficient methods.

Reviewer 2 Report

1. The abstract must be written highlighting the actual research results received.

2. The problem this reach is trying to solve is not clear from the introduction.

3. The main research contribution is missing in the introduction.

4. The introduction is too long, authors are suggested to only keep the import information and if required a separate section named background can be introduced for keeping the additional information.

5. All the parameters used in the equations must be elaborated in the text.

6. No explanation is provided about the dataset used for experimentation and validating the results.

7. The proposed work must be compared with some state-of-the-art works from 2021-2022 for a better understanding of the capabilities of the proposed work.

8. No future work information is presented in the conclusion.

Reviewer 3 Report

According to the topic raised in this manuscript and following the principles of writing. The manuscript is accepted.

Reviewer 4 Report

This paper develops an anomaly detection approach on financial time series based on principal component analysis and neural networks. Anomaly detection is achieved in two steps. The first step aims at identifying the contaminated time series. The second step focuses on the localization of the anomaly. Simulation studies are conducted for model verification. While I agree that the authors conducted comprehensive work, I have some major concerns listed as follows:

1. The organization of the Introduction section and Section 4 is hard to follow. The structure needs to be significantly improved for better logical coherence.

2. The motivation is not clearly stated, in terms of the current organization of Introduction section.

3. A major limitation is that this paper builds a model but validates it only using simulated data set. No real financial time series data is used.

4. It seems that the data size is very large in the simulation study. This large data requirement limits the paper’s practicability.

5. The authors claimed that they successfully identified “non-extrema anomaly” (such as in Figure 9). In practice, what is real “anomaly”? From the figure it just looks like normal data. Not sure why they should be considered as anomaly in practice.

6.  How is the model performance’s sensitivity to the magnitude of an anomaly?

7. Are L and A vectors? Or Matrices as the authors claimed?

Round 2

Reviewer 2 Report

1. The abstract does not show the experimental result of the research done.

2. A through proof reading is suggested as there are still a few typos.

Reviewer 4 Report

The authors have addressed most of my previous comments, I do not have additional comments.

Author Response

No additional comments.